# Width evolution of channel belts as a random walk

**Jens M. Turowski**[1,2]**, Fergus McNab**[1]**, Aaron Bufe**[1,3]**, and Stefanie Tofelde**[4]

[1]Helmholtz Zentrum Potsdam, GeoForschungsZentrum (GFZ) Potsdam, Potsdam, Germany
[2]State Key Laboratory of Hydroscience and Engineering, Tsinghua University, Beijing, China
[3]Department of Earth and Environmental Sciences, Ludwig Maximilian University Munich, Munich, Germany
[4]Institute of Geological Sciences, Freie Universität Berlin, Berlin, Germany

**Correspondence:** Jens M. Turowski (jens.turowski@gfz-potsdam.de)

**Abstract.** Channel belts form through the mobilization and deposition of sediments during the lateral migration of rivers. Channel-belt width and its temporal evolution are important for the hydraulics, hydrology, and ecology of landscapes, as well as for human activities such as farming, protecting infrastructure, and natural hazard mitigation. Yet, we currently lack a comprehensive theoretical description of the width evolution of channel belts. Here, we explore the predictions of a physics-based model of channel-belt width for the transient evolution of channel belts. The model applies to laterally unconfined channel belts in foreland areas as well as to laterally confined channel belts in mountain settings (here, channel-belt width equals valley floor width). The model builds on the assumption that the switching of direction of a laterally migrating channel can be described by a Poisson process, with a constant rate parameter related to channel hydraulics. As such, the lateral migration of the channel can be viewed as a nonstandard one-dimensional random walk. In other words, at each river cross section the river randomly moves either to the left or right at a given time. The model predicts three phases in the growth of channel belts. First, before the channel switches direction for the first time, the channel belt grows linearly. Second, as long as the current width is smaller than the steady-state width, growth follows an exponential curve on average. Finally, there is a drift phase, in which the channel-belt width grows with the square root of time. We exploit the properties of random walks to obtain equations for the distance from a channel that is unlikely to be inundated in a given time interval (law of the iterated logarithm), distributions of times the channel requires to return to its origin and to first arrive at a given position away from the origin, and the mean lateral drift speed of steady-state channel belts. All of the equations can be directly framed in terms of the channel's hydraulic properties, in particular its lateral transport capacity that quantifies the amount of material that the river can move in lateral migration per unit time and channel length. The distribution of sediment age within the channel belt is equivalent to the distribution of times to return to the origin, which has a right-hand tail that follows a power-law scaling with an exponent of $-3/2$. As such, the mean and variance of ages of sediment deposits in the channel belt do not converge to stable values over time but depend on the time since the formation of the channel belt. This result has implications for storage times and chemical alteration of floodplain sediments, as well as the interpretation of measured sediment ages. Model predictions compare well to data of sediment age distributions measured at field sites and the temporal evolution of channel belts observed in flume experiments. Both comparisons indicate that a random walk approach adequately describes the lateral migration of channels and the formation of channel belts. The theoretical description of the temporal evolution of channel-belt width developed herein can be used for predictions, for example, in hazard mitigation and stream restoration, and to invert fluvial strata for ambient hydraulic conditions. Further, it may serve to connect models designed for either geological or process timescales.

## 1  Introduction

Rivers migrate laterally. Lateral river migration establishes the channel belt, which is defined as the corridor of channel migration formed during one river avulsion cycle (Bridge and Leeder, 1979; Nyberg et al., 2023). Channel belts include the river channel and active bars, levees and abandoned channels, and other areas affected by the river during floods or migration (Fig. 1a) (Nyberg et al., 2023). They can be represented by the planform area that the river has interacted with since its last avulsion, and they can be either unconfined, for example in foreland areas, or confined, for example by valley walls in mountain regions (Fig. 1a and b) (e.g. Howard, 1996; Limaye, 2020; Turowski et al., 2024). Channel belts affect catchment hydrology, host aquifers and hydrocarbon deposits (e.g. Anderson et al., 1999; Blum et al., 2013; Bridge, 2001), and present a key location for organic carbon storage and alteration (e.g. Repasch et al., 2021). During lateral migration, rivers deposit sediment or erode previously deposited sediment, thereby affecting chemical weathering, nutrient transport, and ecology (e.g. Fotherby, 2009; Jonell et al., 2018; May et al., 2013; Miller, 1995; Naiman et al., 2010; Schumm and Lichty, 1963; Torres et al., 2017). Further, the exchange of sediment during lateral channel migration determines the distribution of ages of the sediment stored at and near the surface along rivers, with implications for landscape dynamics, the interpretation of fluvial stratigraphy, and nutrient cycles (e.g. Bradley and Tucker, 2013; Galeazzi et al., 2021; Marr et al., 2000; Pizzuto et al., 2017; Scheingross et al., 2021). Landforms such as backswamps or oxbow lakes, which are specific to channel belts, often host unique ecological communities that depend on regular floods (e.g. Baley, 1991; Junk et al., 1989; Meitzen et al., 2018). Finally, lateral bank erosion is an important natural hazard that can destroy agricultural areas and infrastructure (e.g. Badoux et al., 2014; Best, 2019). All of the mentioned effects make channel belts an important component of fluvial response to environmental change (e.g. Hajek and Straub, 2017). As such, channel belts record a river's past activity and can be used as archives for Earth's history on the timescale of hundreds to thousands of years (e.g. Allen, 1978; Bridge and Leeder, 1979; Galeazzi et al., 2021).

The long-term dynamics of channel belts have been studied separately for meandering (e.g. Camporeale et al., 2005; Greenberg and Ganti, 2024; van de Lageweg et al., 2013) and braided rivers (e.g. Bertoldi et al., 2009; Limaye, 2020). Researchers have largely focused on channel characteristics and statistics, their temporal evolution, and approach to a steady state. For meandering rivers, these have typically included the linear and curvilinear wavelength, the curvature of the channel, and the role of meander cuts-offs in reaching and maintaining a steady state (e.g. Camporeale et al., 2005; Howard, 1996). For braided rivers, they have typically included braiding indices and planform patterns (e.g. Bertoldi et al., 2009; Egozi and Ashmore, 2009). In comparison to these statistics describing the channels within the channel belt, the belt width has received little attention. Dong and Goudge (2022) suggested that channel belt width systematically decreases with the number of channels in the river system. As such, the belt width of braided channels is lower than that of meandering channels. Greenberg et al. (2024) found that channel-belt area scales with floodplain reworking timescales. Reworking timescales monotonically increase as water partitions into fewer active channel threads and as channels become more sinuous, and thus they vary between river systems with different planform types. Studying models of meandering rivers, Camporeale et al. (2005) concluded that one timescale and one length scale are sufficient to explain steady-state characteristics of channel belts regardless of the hydrodynamic complexity of the underlying model. They suggested that channel-belt width scales with the meandering wavelength, which in turn scales with flow depth. A qualitative comparison to natural channels was favourable. Limaye (2020) postulated that the channel-belt width of braided rivers scales with channel width. Using flume experiments, he showed that both channel width and belt width follow a similar scaling relationship with discharge. Turowski et al. (2024) developed a steady-state model for confined and unconfined channel-belt width under the assumption that switches in the direction of lateral channel migration are based on a random process with a uniform mean rate of switching in time. In their model, the unconfined steady-state channel-belt width linearly depends on flow depth. The steady-state width of confined channel belt (i.e. the valley floor width) is reduced relative to unconfined channel belts due to lateral input of sediments from adjacent valley walls.

The temporal evolution of channel-belt width has so far hardly been explored. Limaye (2020) identified three phases of channel-belt growth in his experiments, co-occurring with distinct phases of meandering or braiding. In the first phase, the channel established a graded geometry from the initial imposed boundary condition. In the second phase, the channel belt grew rapidly, while in the third phase, it reduced its growth rate. When compared in a dimensionless framework, the switches between phases occurred at the same dimensionless time for different experimental conditions. Wickert et al. (2013) and Bufe et al. (2019) observed an exponential approach to the steady-state width in experiments when tracking the increase in the area visited by the channel over time. Howard (1996) found that the width of the channel belts in a model of meandering channels grows logarithmically over time. Hancock and Anderson (2002) suggested that the initial rapid widening rate of a channel belt and its subsequent decrease are due to the declining probability of the channel being located at the belt boundary as the belt widens. This notion was regularly picked up in later work (e.g. Malatesta et al., 2017; Martin et al., 2011) and has led to steady-state descriptions of valley width (Tofelde et al., 2022; Turowski et al., 2024). Equations relating the growth evolution of con-

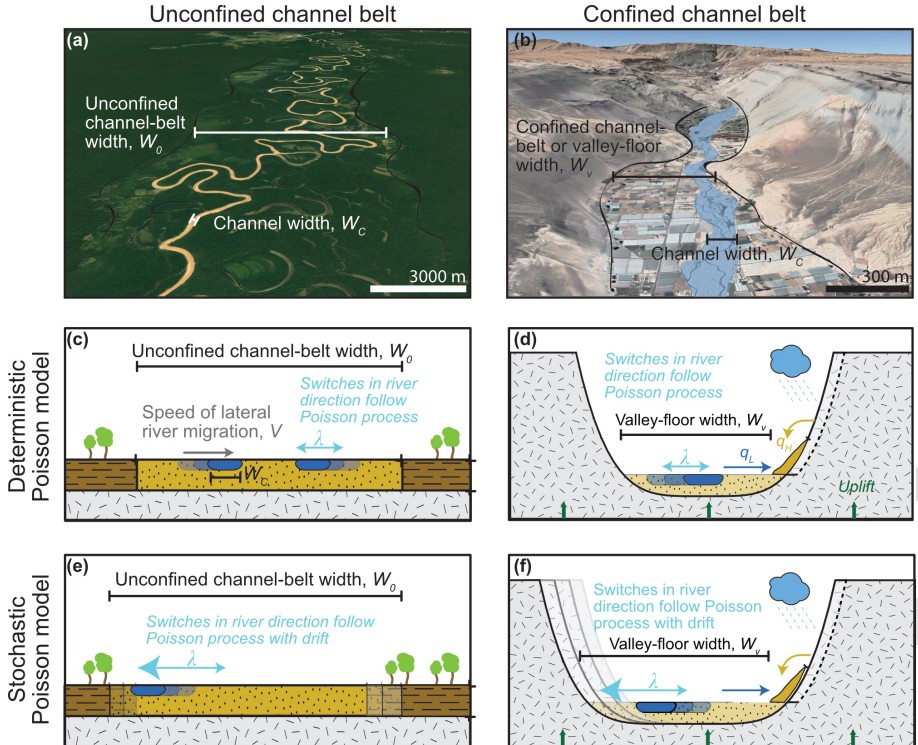

**Figure 1.** Schematic illustration of the model concept. **(a)** Unconfined channel belt of the Juruá River, Brazil (6.75° S, 70.30° W; map data: © Google Earth. Imagery: © 2024 Maxar Technologies, Airbus). **(b)** Confined channel belt of the San Jose River, Chile (18.58° S, 69.97° W; map data: © Google Earth. Imagery: © 2024 Maxar Technologies, Airbus). **(c, d)** The channel switches the direction of motion after a certain timescale. It thus evolves to a steady-state width that does not change over time. In the stochastic Poisson model **(e, f)**, the switching timescale is a random number. As such, the channel may migrate beyond the channel-belt limits **(e)** or erode the valley walls even after reaching the steady-state width. The resulting migration can lead to a lateral drift of the unconfined or confined channel belt.

fined and unconfined channel belts to the hydraulic conditions in the channel are currently not available. Yet, they could be useful for diverse topics. For example, they could be used as forward models for making predictions related to flood hazard assessment and stream restoration or as inverse models to obtain paleo-hydraulic conditions from fluvial stratigraphy and depositional sequences. Further, they could provide a framework to interpret data from natural rivers with regard to nutrient cycling, channel–floodplain interactions, and ecology.

Turowski et al. (2024) described lateral channel migration as a Poisson process, in which the switches in direction occur randomly in time at a constant mean rate. They subsequently focused on the mean behaviour of the model and proceeded to derive equations for the steady-state width of unconfined and confined channel belts. Here, we explore the predictions of their model concept for the transient approach of channel belts to their steady-state width and the consequences of a stochastic formulation for channel-belt dynamics. Specifically, we derive analytical equations describing the temporal evolution and the bounds of channel belts, their average lateral drift once they have reached a steady state, and the sediment residence time distribution, which is equivalent to the distribution of sediment ages. Analytical results are benchmarked with stochastic numerical simulations. We compare the model results to data from two flume experiments (Bufe et al., 2016a, 2019) and sediment age distributions from three field sites (Everitt, 1968; Huffman et al., 2022; Skalak and Pizzuto, 2010).

## 2 Theoretical developments

In this section, we will briefly summarize the valley width model by Turowski et al. (2024) (Sect. 2.1). Afterwards, we outline the basis of the stochastic model approach used herein (Sect. 2.2). Then, we derive equations for the temporal evolution of channel belts while approaching a steady state and their lateral drift speed once they have reached steady state (Sect. 2.3), the limits of the channel-belt bounds (Sect. 2.4), the first passage distribution (Sect. 2.5), and the age distribution of sediment (Sect. 2.6).

### 2.1 Summary of the steady-state model

Building on earlier work (e.g. Bufe et al., 2019; Martin et al., 2011; Tofelde et al., 2022), Turowski et al. (2024) de-

veloped a model for the steady-state width of fluvial valleys (Fig. 1), which includes predictions for confined and unconfined channel belts. In the model, each cross section contains a single channel, which is treated as if it moves independently from those upstream and downstream. River channels are assumed to move laterally by bank erosion and deposition. The channel belt widens when the river crosses beyond the previous channel-belt boundaries (Fig. 1). The lateral channel migration speed $V$ [$L\,T^{-1}$] is equal to the ratio of the lateral transport capacity $q_L$ [$L^2\,T^{-1}$] and the bank height in the direction of motion $H_+$ [L], where $q_L$ quantifies the amount of material that the river can move in the lateral direction per unit time and channel length (Bufe et al., 2019):

$$V = \frac{q_L}{H_+}. \tag{1}$$

The lateral transport capacity can be treated as a constant for a given set of boundary conditions including water discharge, upstream sediment supply, and granulometry (Bufe et al., 2019). Turowski et al. (2024) viewed switches in the direction of lateral motion of the channel as stochastic events. These are assumed to be independent and identically distributed, with a constant mean event rate per unit time, $\lambda$ [$T^{-1}$], and can therefore be described by a Poisson process. The mean rate of switching $\lambda$ is proportional to the ratio of the lateral transport capacity $q_L$ and the square of the flow depth $h$ [L] (Turowski et al., 2024):

$$\lambda = k\frac{q_L}{h^2}, \tag{2}$$

where $k$ [–] is a dimensionless constant. We can define an effective switching timescale as a constant timescale that leads to the same steady-state width as is obtained from a fully stochastic model. The effective switching timescale $\Delta T$ [T] is inversely proportional to $\lambda$:

$$\Delta T = \frac{c}{\lambda}, \tag{3}$$

where $c$ [–] is a dimensionless constant of order 1. Integrating over the distance travelled laterally by the channel within $\Delta T$ yields an equation for the unconfined channel-belt width $W_0$ [L] (see Turowski et al., 2024, for details):

$$W_0 = \int_0^{\Delta T} V\,dt + W_C = k_0 h + W_C. \tag{4}$$

Here, $k_0 = c/k$ [–] is a dimensionless constant, $W_C$ [L] is the channel width, and $t$ [T] is time. To arrive at the final equality in Eq. (4), we assumed that in an unconfined channel belt that is neither incising nor aggrading, the bank height in the direction of motion, $H_+$, is equal to the flow depth, $h$ (see Turowski et al., 2024). In river valleys, the channel belt or valley floor is narrower than $W_0$ due to uplift or lateral supply of sediment from hillslopes, and the steady-state valley floor

width $W_V$ [L] can be described by the equation (Turowski et al., 2024)

$$W_V = \left(\frac{q_L - q_H}{U}\right)\ln\left\{1 + \frac{U(W_0 - W_C)}{q_L}\right\} + W_C. \tag{5}$$

Here, $q_H$ [$L^2\,T^{-1}$] is the lateral supply rate of hillslope sediment per unit channel length, and $U$ [$L\,T^{-1}$] is the uplift rate. The valley floor width $W_V$ is distinguished from the confined channel-belt width by explicitly accounting for the effects of uplift and lateral sediment supply. Equation (5) predicts that river valleys reach a steady-state width that depends on five input parameters (flow depth $h$, channel width $W_C$, uplift rate $U$, lateral transport capacity $q_L$, and lateral hillslope sediment supply $q_H$) and one constant ($k_0$) that needs to be determined from observations. Steady-state valley width is reached when the system achieves a balance between local sediment input from hillslopes by uplift, on the one hand, and the removal of sediment by the river on the other hand.

In summary, in their model, Turowski et al. (2024) assume that the switches in river direction follow a Poisson process and unconfined channel belts evolve to a steady-state width determined by flow depth and channel width (Eq. 4). Fluvial valleys can attain a maximum steady-state width that corresponds to the unconfined channel-belt width $W_0$. They are narrower than this unconfined width if they are affected by uplift or lateral hillslope sediment supply (Eq. 5). We call this model the deterministic Poisson model hereafter.

## 2.2 The stochastic Poisson model

In order to investigate the temporal evolution of channel-belt width, we further develop the previous model of Turowski et al. (2024). Instead of assuming the channel switches with a constant characteristic timescale, the effective switching timescale $\Delta T$ (Eq. 3), we now explore the consequences of a random switching timescale. This consideration allows us to observe the temporal behaviour of the random-walk model for lateral river migration. We call this model the "stochastic Poisson model" hereafter. In a Poisson process, the probability mass function (PMF) that $n$ [–] events (in this case, channel switches) occur within a time of length $\Delta t$ [T] is given by

$$\text{PMF}_{\text{Poisson}} = \frac{(\lambda\Delta t)^n e^{-\lambda\Delta t}}{n!}. \tag{6}$$

Both the expected number of events and their variance are given by $\lambda\Delta t$ [–]. For the derivations within this paper, we use the idea that the lateral motion of the river channel across the floodplain, in the model concept of a Poisson process, can be viewed as a nonstandard one-dimensional random walk. The channel alternates between steps to the left and to the right within the cross section, thus switching direction after every step. The step length is not a constant but a stochastic parameter equal to the waiting times between individual

switching events multiplied by lateral migration speed. In a Poisson process, the waiting times $T_W$ [T] between events are exponentially distributed with a mean waiting time of $1/\lambda$, a variance of $1/\lambda^2$, and a probability density function (PDF) given by

$$\text{PDF}_{T_W} = \lambda e^{-\lambda T_W} . \tag{7}$$

Similarly, for constant migration speed $V$ [L T$^{-1}$], the PDF of the length of steps $\Delta x = V \Delta t$ [L] is given by

$$\text{PDF}_{\Delta x} = \frac{\lambda}{V} e^{\frac{-\lambda}{V} \Delta x} . \tag{8}$$

In the following, we will first derive an equation for the approach to the steady-state width using the deterministic Poisson model (Turowski et al., 2024) and then use the mathematics of random walks to explore the effects of stochasticity on the channel belt's temporal evolution. Finally, we investigate the distribution of floodplain ages.

## 2.3 Temporal evolution of the channel-belt width

### 2.3.1 Approach to steady state in the deterministic Poisson model

We first consider the evolution of the channel belt in an unconfined setting. Consider a river channel moving laterally with speed $V$. The channel belt widens when the river is located at and moves into the channel-belt boundary. In contrast, if the river is not located at the boundary or moves away from it, the channel-belt width remains unchanged. At any given time, widening can be observed with a probability $P$ [–], which is equal to the fraction of the time the river spends widening the valley (e.g. Hancock and Anderson, 2002; Tofelde et al., 2022). The temporal evolution of channel-belt width $W$ [L] is then governed by the differential equation (Tofelde et al., 2022):

$$\frac{dW}{dt} = PV . \tag{9}$$

Motion in either direction is equally likely, and, for a given set of hydraulic, tectonic, and sedimentological boundary conditions, $V$ can be considered a constant (Bufe et al., 2019; Turowski et al., 2024). In a transient phase, before the steady-state width is reached, the probability of the river not widening, i.e. $1 - P$, the channel belt is equal to the ratio of the current $W$ [L] and the maximum $W_0$ [L] channel-belt width (Tofelde et al., 2022). Channel width $W_C$ provides a starting point and needs to be subtracted. Thus, $P$ is given by (Turowski et al., 2024)

$$P = 1 - \frac{W - W_C}{W_0 - W_C} = \frac{W_0 - W}{W_0 - W_C} . \tag{10}$$

The speed of lateral motion is equal to the ratio of the lateral transport capacity and the height of the bank in the direction of motion $H_+$ (Eq. 1). Combining Eqs. (1), (9), and (10),

we obtain a differential equation for channel-belt evolution:

$$\frac{dW}{dt} = \frac{W_0 - W}{W_0 - W_C} \frac{q_L}{H_+} . \tag{11}$$

Solving Eq. (11) and applying the boundary condition that channel-belt width $W$ is equal to $W_C$ at time $t = 0$, we obtain

$$W(t) = W_0 - (W_0 - W_C) \exp\left\{ -\frac{t}{\tau} \right\} + W_C . \tag{12}$$

Here, $\tau$ is the governing timescale, which can be interpreted as a response timescale to an external perturbation (cf. Tofelde et al., 2021). It is given by

$$\tau = (W_0 - W_C) \frac{H_+}{q_L} . \tag{13}$$

In the unconfined case, $H_+$ is equal to flow depth $h$, and substituting Eqs. (1) and (2) into Eq. (11), we find that $\tau$ is equal to the effective switching timescale $\Delta T$ (see Eqs. 3 and 4):

$$\tau = \frac{c}{\lambda} = \Delta T . \tag{14}$$

We can use a similar approach to describe the evolution of a channel belt that is confined by valley walls when considering that at the valley walls, the lateral migration of the river slows down (cf. Eq. 1). If the valley walls are made of alluvium, the bank height $H_+$ in Eq. (9) is equal to the height of the valley wall $H_W$ [L] and Eq. (1) can be used as before. However, we need to adjust Eq. (10), defining an equivalent probability $P_{\text{confined}}$ for a confined channel belt. The distance $d$ [L] is the length that a channel moves on average across the valley floor in the effective time $\Delta T$ [T] between two events of switching the direction of motion. This distance $d$ is the sum of the distance covered at higher speed $V$ when moving in the floodplain and the distance covered when moving at lower speed $v$ [L T$^{-1}$] when cutting into the valley walls (cf. Tofelde et al., 2022):

$$d = V(1 - P_{\text{confined}})\Delta T + v P_{\text{confined}}\Delta T . \tag{15}$$

For the unconfined channel belt, we know that

$$V \Delta T = W_0 - W_C . \tag{16}$$

Using Eq. (16) to eliminate $\Delta T$ in Eq. (15), and noting that $d$ corresponds to the current width $W - W_C$, we obtain

$$\begin{aligned} P_{\text{confined}} &= \frac{W_0 - W}{(W_0 - W_C)\left(1 - \frac{v}{V}\right)} \\ &= \frac{W_0 - W}{(W_0 - W_C)\left(1 - \frac{H_W}{h}\right)} . \end{aligned} \tag{17}$$

Here, we use Eq. (1) to substitute for $V$ and $v$ using $H_+ = h$ and $H_+ = H_W$, respectively. Note that in the assumption behind Eqs. (15) to (17), $P_{\text{confined}}$ for a confined valley (Eq. 17)

reduces to $P$ for an unconfined floodplain (Eq. 8) for $v = 0$ or $H_W = 0$ (rather than $v = V$ or $H_W = h$). This arises from Eq. (15), which yields $d = V \Delta T$ for $v = V$, rendering $P_{confined}$ meaningless. Substituting Eq. (17) into Eq. (9) and integrating again yields Eq. (12) with a different governing timescale $\tau$ given by

$$\tau = \frac{(W_0 - W_C)(H_W - h)}{q_L} = \left(\frac{H_W}{h} - 1\right)\frac{c}{\lambda}. \tag{18}$$

### 2.3.2 Channel-belt evolution in the stochastic Poisson model

As in Sect. 2.3.1, we first consider the evolution of an unconfined channel belt. In the deterministic Poisson model, we obtained an exponential approach to the steady-state width (Eq. 12) (Sect. 2.3.1). In the stochastic Poisson model, we can distinguish three different phases in the growth of the channel-belt width over time. In the first phase, before the first switch in direction occurs, width increases linearly in time. In this phase, the growth rate is determined by the speed of lateral channel migration, $V$ in the unconfined case and $v$ in the confined case (see Eq. 1 and Sect. 2.3.1). In the second phase, before reaching the steady-state width, the channel-belt width grows exponentially on average. This average exponential growth can be described by the same equation (Eq. 12) that has been derived for the deterministic Poisson model (see Sect. 2.3.1). In the third phase, which starts approximately when the width for the first time reaches the steady-state width, stochastic drift dominates. Stochastic drift arises because, due to the random motion of the channel, there is always a finite probability of widening the belt even after the steady-state width has been reached. We already have equations for the linear (Eq. 1) and exponential (Eq. 12) phase. In the following, we will fully exploit the stochastic properties of the model concept. In several of our considerations in this and the following sections, we use the central limit theorem, which states that the sum $X$ of $n$ stochastic variables with mean $\mu$ and variance $\sigma^2$ is normally distributed with mean $n\mu$ and variance $n\sigma^2$ if $n$ is sufficiently large. In addition, we use the result that the sum or difference of two normally distributed parameters with means $\mu_1$ and $\mu_2$ and equal variance $\sigma^2$ follows a normal distribution with mean $\mu_1 \pm \mu_2$ and variance $2\sigma^2$.

First, we will derive an equation for widening during the drift phase using the evolution of random walks in the limit of a large number of steps. In this case, we can apply the central limit theorem. Thus, the PDF of the location of the channel can then be described by a normal distribution. In a random walk, the width of this normal distribution increases with the square root of its variance $\text{VAR}_{UCB}$ [$L^2$], where the subscript stands for "unconfined channel belt" (e.g. Lawler and Limic, 2010):

$$W_{Drift} = \sqrt{\text{VAR}_{UCB}} + W_C. \tag{19}$$

To find an equation for the variance, we will use the concept of a random walk, making steps in alternating directions with exponentially distributed step length. We consider $m$ pairs of a total of $n$ steps, where each of the $n$ steps covers an average distance of $V/\lambda$. The difference of two consecutive identically exponentially distributed steps in opposite directions is described by the Laplace distribution with zero mean and variance $2V^2/\lambda^2$, with the PDF

$$\text{PDF}_L = \frac{\lambda}{2V} e^{\frac{-\lambda}{V}|x|}. \tag{20}$$

After each pair of steps, the river is always in a position where it switches direction in the same way, for example from left to right. The switch in the other direction, from negative to positive, also follows Eq. (20). In the limit of large $m$, the position of the river is given by the sum of the positions of many step pairs. The central limit theorem applies, and the normal approximation gives the distribution of locations where the river switches either from positive to negative or vice versa, with zero mean and a variance of $2mV^2/\lambda^2 = nV^2/\lambda^2$. Finally, the channel-belt width is the difference of the switching position on either side, so the final variance needs to be multiplied by a factor of 2. Applying the law of large numbers, the distance covered in the sum of all steps is equal to the number of steps times the average step length $V/\lambda$. The average time of each step is the mean waiting time $1/\lambda$, and so we can write $n = \lambda t$:

$$\text{VAR}_{UCB} = 2n\frac{V^2}{\lambda^2} = 2\frac{t}{\lambda}V^2 = \frac{2}{k}q_L t. \tag{21}$$

Thus, we obtain the width increase due to drift from Eqs. (19) and (21) as

$$W_{Drift}(t) = \sqrt{\frac{2}{k}q_L t} + W_C. \tag{22}$$

For a confined channel belt, during the time the river incises into the confining walls, the speed of widening drops to $q_L/H_W$, where $H_W$ is the height of the confining wall, while it remains at $q_L/h$, as before when the river moves laterally within the channel belt. The average speed of motion is given by the geometric average of the two speeds, $\overline{V}$:

$$\overline{V} = \sqrt{vV} = \sqrt{\frac{h}{H_W}}V. \tag{23}$$

We obtain the variance by replacing $V$ by $\overline{V}$ in Eq. (22), giving the variance $\text{VAR}_{CCB}$ for a confined channel belt:

$$\text{VAR}_{CCB} = 2t\overline{V}^2/\lambda = 2q_L th/kH_W. \tag{24}$$

As before, the width during the drift phase evolves as the square root of the variance, giving

$$W_{Drift}(t) = \sqrt{2\frac{t}{\lambda}\overline{V}} + W_C = \sqrt{\frac{2}{k}\frac{h}{H_W}q_L t} + W_C. \tag{25}$$

### 2.3.3 Drift speed of channel belts and dimensionless scaling factor of the mean switching timescale

During the drift phase, the channel belt widens laterally, increasing the area that has been reworked by the channel with the square root of time (Eq. 25). Yet, growth on one side of the channel belt makes it less likely that the channel will move close to the other side. As such, parts of the channel belt may be abandoned and, for example reclaimed by vegetation (Fig. 1e). Similarly, in the case of a vertically incising river, the channel-belt width can stay at the steady-state value $W_V$ (Eq. 5), while the entire belt is moving laterally, and uplift converts old parts of the channel belt to fluvial terraces. Here, we consider the case in which the channel belt keeps its width constant at the steady-state width because any acquisition of area of the belt due to lateral motion on one side leads to the abandonment of an equivalent area on the other side. In this case, instead of widening, during the drift phase, the entire belt drifts laterally. We will now derive an equation for the average drift speed in this case. The average drifted distance in one direction, $X_{\text{Drift}}$, is equal to the square root of the variance, as before (cf. Eq. 19). Because we consider a distance, rather than the width, it is smaller by a factor of 2 in comparison to Eq. (25), giving

$$X_{\text{Drift}}(t) = \sqrt{\frac{1}{k} \frac{h}{H_W} q_L t} . \tag{26}$$

The derivative of Eq. (26) with respect to time, evaluated at the time when the valley reaches its steady-state width, $T_{\text{SS}}$ [T], gives the drift speed $V_{\text{Drift}}$ [L T$^{-1}$]:

$$V_{\text{Drift}} = \frac{1}{2} \sqrt{\frac{1}{k} \frac{h}{H_W} \frac{q_L}{T_{\text{SS}}}} . \tag{27}$$

At time $T_{\text{SS}}$, $X_{\text{Drift}}$ is equal to the steady-state width $W_0$, and we can use Eq. (26) to obtain

$$T_{\text{SS}} = k \frac{H_W}{h} \frac{(W_0 - W_C)^2}{2 q_L} . \tag{28}$$

Substituting Eq. (28) into Eq. (27) yields

$$V_{\text{Drift}} = \frac{1}{\sqrt{2} k} \frac{h}{H_W} \frac{q_L}{(W_0 - W_C)} . \tag{29}$$

We can use Eq. (29) to arrive at a further result and calculate the constant of proportionality $c$ between the switching timescale $\Delta T$ and the rate constant $\lambda$ (Eq. 3). The ratio of the drift speed $V_{\text{Drift}}$ and the lateral migration speed of the channel $V$ is the same as the fraction of time that the river spends widening the channel belt. This is equal to the area under a normal distribution outside 1 standard deviation from the mean, $V_{\text{Drift}}/V = 0.3173$. Setting $h/H_W = 1$ and substituting $q_L = V h$, we find

$$\frac{V_{\text{Drift}}}{V} = 0.3173 = \frac{1}{\sqrt{2} k} \frac{h}{(W_0 - W_C)} = \frac{1}{\sqrt{2} c} . \tag{30}$$

Equation (30) therefore yields $c = 2.2285$.

### 2.4 Channel-belt limits

We can use the properties of random walks to make a statement about the distance beyond which the river will rarely migrate over a given timescale. Knowledge of this distance may be useful to delineate zones for building or to assess in which areas the river is likely (or not) to interact with its surrounding, for example, by reworking sediment or evacuating erosion and weathering products. In random walks, this distance is described by the law of the iterated logarithm (e.g. Kolmogoroff, 1929), which is a limit theorem that sits in between the central limit theorem and the law of large numbers. In the limit of a large number of steps, this law provides an envelope to the area that the river almost surely will not leave in its stochastic motion. Consider the sum $S$ over the distance travelled in $n$ steps over dimensionless time $t^*$, which is a dimensionless stochastic variable with zero mean. The law of the iterated logarithm gives an upper and lower bound for this sum with the equation

$$S = \pm \sqrt{2 t^* \ln\{\ln\{t^*\}\}} . \tag{31}$$

Here, ln denotes the natural logarithm, and the plus and minus give the upper and lower bound, respectively. We define the dimensionless step length $s = \lambda \Delta x / V$. This step length is a stochastic variable that is exponentially distributed with a mean of zero and variance equal to 1 (compare to Eq. 7). Because the random walk has to be symmetric for Eq. (31) to apply, we consider the sum $S$ of $m = n/2$ pairs of steps, distributed according to the Laplace distribution (Eq. 20). Normalizing with the square root of the variance of the Laplace distribution, the dimensional distance is then given by $X = \sqrt{2} S V / \lambda$. This is the distance from the origin that the channel will almost surely not cross within timescale $t$. The dimensionless time is given as $t^* = 2 V t / h$, where the factor of 2 accounts for the pairs of steps. Putting everything together and adding half of the channel width, we obtain

$$
\begin{aligned}
X(t) &= \sqrt{2} \frac{SV}{\lambda} + \frac{W_C}{2} \\
&= \pm 2 \frac{h}{k} \sqrt{2 \frac{\lambda t}{k} \ln\left\{\ln\left\{2 \frac{\lambda t}{k}\right\}\right\}} + \frac{W_C}{2} .
\end{aligned}
\tag{32}
$$

### 2.5 First passage time distribution

We can derive another result that may be useful for planning and hazard mitigation purposes over long timescales. The first passage time distribution (e.g. Redner, 2001) is the distribution of times until the channel reaches a point that is located a distance $b$ [L] from the channel's original location for the first time. This distribution can be used, for example, to calculate a lifetime distribution of structures located a distance $b$ from the river. In random walks, the first passage time distribution is given by a Lévy distribution. The distribution

PDF$_{\text{FP,R}}$ of times $T_{\text{FP}}$ [T] is given by

$$\text{PDF}_{\text{FP,R}}(T_{\text{FP}}) = \frac{|b|}{\sqrt{2\pi \frac{h}{H_{\text{W}}} \frac{q_{\text{L}}}{k} T_{\text{FP}}^3}} \exp\left\{ \frac{-b^2}{2\frac{h}{H_{\text{W}}} \frac{q_{\text{L}}}{k} T_{\text{FP}}} \right\}. \quad (33)$$

## 2.6   Sediment residence time distribution

The probability distribution of residence times may be useful to calculate the age distribution of sediments. This is relevant, for example, for understanding weathering rates in river deposits or transfer times of sediment and carbon to the ocean (e.g. Repasch et al., 2021; Scheingross et al., 2019; Tofelde et al., 2021). The residence time distribution differs from the first passage distribution (Sect. 2.5) but can be derived from it. We start with a single step outward. The migrated distance $\Delta x$ until the channel switches direction is then given by the exponential distribution (Eq. 8). We can then use the first passage distribution (Eq. 33) for the time to return to the origin by again migrating a distance $b = \Delta x$. Finally, we need to account for all possible $\Delta x$ values in the initial step. Assuming that the first step has to erode into the valley walls, the distribution PDF$_{\text{RT}}$ for the time needed to return to the origin $T_{\text{R}}$ [T] is then given by

$$\text{PDF}_{\text{RT}}(T_{\text{R}}) = \int_0^{\frac{h}{H_{\text{W}}}Vt} \frac{\lambda}{\frac{h}{H_{\text{W}}}V} \exp\left\{ \frac{-\lambda}{V} \Delta x \right\}$$

$$\times \frac{|\Delta x|}{\sqrt{2\pi \frac{h}{H_{\text{W}}} \frac{q_{\text{L}}}{k}\left( T_{\text{R}} - \frac{\Delta x}{\frac{h}{H_{\text{W}}}V} \right)^3}} \quad (34)$$

$$\times \exp\left\{ \frac{-\Delta x^2}{2\frac{h}{H_{\text{W}}} \frac{q_{\text{L}}}{k}\left( T_{\text{R}} - \frac{\Delta x}{\frac{h}{H_{\text{W}}}V} \right)} \right\} \text{d}\Delta x.$$

Unfortunately, Eq. (34) does not yield an analytical solution, but it can be solved numerically. We can find an analytical limit for the right-hand tail when $T_{\text{R}}$ is large. Then, the integral reduces to

$$\text{PDF}_{\text{RT}}(T_{\text{R}} \gg 0) = \int_0^{\infty} \frac{\lambda}{\frac{h}{H_{\text{W}}}V} \frac{|\Delta x|}{\sqrt{2\pi \frac{h}{H_{\text{W}}} \frac{q_{\text{L}}}{k}(T_{\text{R}})^3}}$$

$$\times \exp\left\{ \frac{-\lambda}{V} \Delta x \right\} \text{d}\Delta x \quad (35)$$

$$= \frac{\lambda}{\sqrt{2\pi}} \left( \frac{h}{H_{\text{W}}} \lambda T_{\text{R}} \right)^{-3/2}.$$

We suggest an analytical approximation for the entire distribution (Eq. 34) by assuming that, for small $T_{\text{R}}$, the PDF approaches a constant. Using this condition together with Eq. (35) and fixing the integral to 1, as required for any dis-

tribution, we obtain the function

$$\text{PDF}_{\text{RT}}(T_{\text{R}}) \approx \frac{1}{\sqrt{2\pi}} \frac{a\frac{h}{H_{\text{W}}}\lambda}{1 + a\left( \frac{h}{H_{\text{W}}} \lambda T_{\text{R}} \right)^{3/2}}, \quad (36\text{a})$$

with

$$a = \left( \frac{3}{2} \right)^3 \left( \frac{3}{2\pi} \right)^{3/2}. \quad (36\text{b})$$

## 3   Testing the stochastic Poisson model

We test the model predictions in two separate ways. First, we use a stochastic random walk model to benchmark the analytical equations (Sect. 3.1) by explicitly using the random properties to calculate the distributions and the mean behaviour. Next to the analytical equations derived so far, this is an independent way of evaluating the stochastic Poisson model. We refer to this approach as the stochastic benchmark and use it to check that the derivations of the analytical equations are correct. Second, we want to test the results with published experimental or field data. A full comparison of all of the results derived herein is beyond the scope of the paper. Instead, we focus on scaling relationships that are indicative of random walks. Thus, we test whether channel belts can be described as a random walk and validate the fundamental modelling assumptions and the approach that we used to derive the analytical equations. Two results are particularly suitable for this test. First, published distributions of floodplain sediment ages (Everitt, 1968; Huffman et al., 2022; Skalak and Pizzuto, 2010) (Sect. 3.2) allow us to measure the sediment residence time distribution and test the prediction of a $-3/2$ power-law scaling. Second, the temporal evolution of channel belts in braided-channel experiments (Bufe et al., 2016a, b, 2019) (Sect. 3.3) allows us to extract the average channel-belt width evolution during the drift phase and validate the predicted square root scaling of average width with time during this phase.

## 3.1   Stochastic benchmark calculations

To benchmark the analytical equations, we use a stochastic numerical random walk model, the stochastic benchmark, as an independent evaluation of the stochastic Poisson model to check the analytical equations. The stochastic benchmark builds on the same assumptions used to derive the analytical results but explicitly generates random step lengths of the channel in alternating directions, thereby generating random paths of channel migration. We ran the stochastic benchmark in many iterations, calculated the average behaviour and the corresponding distributions of the properties, and compared them to the analytical results. The analytical equations and the results from the stochastic benchmark are both fully determined and mutually independent, and there is no need to fit any free parameters. The scripts to run and evaluate the

stochastic benchmark and to generate the figures are available in the publication by McNab (2024). Except where otherwise stated, we fixed channel width to zero and all other free model parameters to 1. For each step, the step length was randomly picked from an exponential distribution (Eq. 7), and the lateral position of the channel was tracked by alternately adding or subtracting the obtained step length from the channel's previous position. Channel-belt width was calculated as the difference of the maximum distance that the channel had migrated in the positive and negative directions from the origin up to the time step of interest. In this way, we generated a total of 1000 trajectories of position and channel-belt width, each with a total length of 3000 time steps. We repeated this exercise for ratios of valley depth to channel depth of $H_W/h = 1$, 10, and 100 for unconfined, moderately confined, and highly confined scenarios, respectively. We obtained the average position of the channel for bins spaced logarithmically in time. We used the unconfined width in further simulations to check the drift equation (Eq. 25). For this check, we limited the channel-belt width to the steady-state width by adjusting the one side of the valley in an equal manner when the channel ventured beyond the channel-belt limit on the other side. This procedure results in a valley of fixed width that moves laterally. We measured drift velocity for different steady-state widths by varying the channel depth for different values of the lateral transport capacity, and, as above, for ratios of valley depth to channel depth of $H_W/h = 1$, 10, and 100, as before. These simulations were run for a total of 3000 time steps to ensure statistical convergence. To verify the dimensionless scaling factor $c$ that relates the mean switching time to the rate constant $\lambda$ by $c/\lambda$ (Eq. 3), we compared the unconfined steady-state width for various conditions to flow depth for simulations with $k = 1$ (cf. Eq. 2). To obtain an independent estimate of $W_0$ from the data, we fitted the exponential evolution equation (Eq. 12) to the initial phase of channel-belt widening. To obtain the first passage distribution, we ran 10 000 simulations, each until the walk reached a dimensionless distance of 10 from the starting point. We used the results to construct the first passage distribution. Similarly, to test the distribution of channel belt ages, we ran the random walk simulations until the channel returned to the origin for the first time. We repeated the simulation 10 000 times, for a maximum of 100 000 steps. The time needed to return to the origin in each run was used to construct the distribution of sediment residence times.

## 3.2 Floodplain ages from the field

The $-3/2$ scaling in the distribution for the time needed to return to the origin (Eqs. 34–36) is indicative of random walks, and thus its presence in natural data would be a strong indication that this modelling approach is suitable for describing the dynamics of channel belts. Yet, the controls on sediment ages in natural rivers can be complicated. Depending on the location, sediments may be deposited not only by laterally migrating channels, but also by overbank deposition, tributaries, or other processes such as soil erosion or debris flows. We thus do not expect the sediment age distribution in every river to follow the prediction of our model (Eqs. 34–36). To compare to predictions, we picked three channels with published age distributions that feature conditions close to the assumptions of the model: single-thread channels undisturbed by processes other than fluvial deposition and erosion (e.g. debris flows), without major tributaries in the study area. We digitized floodplain ages published by Everitt (1968) for the Little Missouri River at Watford, North Dakota, USA; by Skalak and Pizzuto (2010) for the South River near Waynesboro, Virginia, USA; and by Huffman et al. (2022) for the Powder River between Moorhead and Broadus, Montana, USA, to compare against the predicted power-law scaling (Eq. 35). In the original study of Skalak and Pizzuto (2010), the cumulative distribution function (CDF) of floodplain ages is shown (their Fig. 8). We estimated the PDF by numerically differentiating the CDF using a centred finite-difference scheme. Note that Skalak and Pizzuto (2010) already reported a power-law scaling with an exponent close to $-3/2$ in their study, while both Everitt (1968) and Huffman et al. (2022) interpreted their data using an exponential function.

## 3.3 Analogue experiments

We further validate the model against experimental data of Bufe et al. (2016b) and Bufe et al. (2019). Primarily, we seek evidence for the drift phase, i.e. the increase in the average channel-belt width with the square root of time in the later parts of the experiments. This would be a strong indication that channel-belt development can be described as a random walk. Bufe et al. (2016b) and Bufe et al. (2019) conducted and analysed experiments on braided alluvial channels in a basin with dimensions of $4.8\,\text{m} \times 3.0\,\text{m} \times 0.6\,\text{m}$ and filled with well-sorted silica sand ($D_{50} = 0.52\,\text{mm}$). Water and sediment were supplied into the basin at a constant rate from the centre of one of the short edges and flowed out of the opposite side of the basin across a weir into a drain. After the start of the experiments, the system evolved into an aggrading braided-channel network. Once the average aggradation rate dropped to below 20 % of the input flux, a flexing metal sheet underneath the basin was used to simulate an uplifting fold. Here, we focus on 25 h of data that were collected before the onset of uplift from Run 5 and on 55 h of data from Run 7, an experiment without uplift (see Bufe et al., 2019, for more details). Water discharge was set to $790\,\text{mL}\,\text{s}^{-1}$ in both experiments, and sediment supply was $15.8\,\text{mL}\,\text{s}^{-1}$ in Run 7 and $2.4\,\text{mL}\,\text{s}^{-1}$ in Run 5. Positions of the channels were tracked at 1 min intervals in overhead images using blue-dyed water and were used to measure the rate at which the area reworked by the channel expanded over time (Bufe et al., 2016b).

## 4   Results

In general, our analytical solutions (Sect. 2) agree well with the stochastic benchmark (Sect. 3.1) (Figs. 2–6), mostly yielding $R^2 > 0.99$ (Table 1). First, we compare the channel location in the stochastic benchmark with the law of the iterated logarithm (Eq. 32) that gives an upper bound on the locations of the channel through time (Fig. 2a) and the expected Gaussian distribution of locations (Fig. 2b). After 3000 steps, no simulated random walk lies outside the predicted bounds (Fig. 2a), and the Gaussian provides a good description of the locations ($R^2 = 0.9962$). Further, we derive the total width of the channel belt in the simulations as the difference between the two outermost points visited by each random walk (Fig. 2c). The temporal evolution of these widths shows all three phases – linear increase, exponential increase, and square root drift – that are expected by the random walk model, and the analytical solutions predict the average behaviour well (Fig. 2c), with $R^2$ values exceeding 0.99 (Table 1).

Keeping the channel-belt width constant at the steady-state channel-belt width, we can measure a displacement of the channel belt with respect to the origin in the stochastic benchmark (Fig. 3a) and calculate an average lateral drift velocity. We find that the average drift velocity is inversely proportional to the steady-state channel-belt width and proportional to the lateral transport capacity (Fig. 3b and c). The relationships agree with the prediction of Eq. (29) (dotted lines in Fig. 3b and c), with $R^2$ values of 0.9999 (Table 1). Further, we find that the steady-state widths of the simulated unconfined random walks increase as a function of the channel depth following a power law with an exponent of $c = 2.2285$ as predicted by Eq. (30) (Fig. 4), with $R^2 = 0.9997$ (Table 1).

The first passage distribution describes the time for the random walk to reach a given distance from the origin and is plotted in Fig. 5a for the stochastic benchmark. Again, the channel does not cross the theoretical bound given by the law of the iterated logarithm (dashed line in Fig. 5a). The mean first passage time in the stochastic benchmark is well fit by Eq. (33) (Fig. 5b), with $R^2 = 0.9991$ (Table 1).

We found a similar correspondence between the stochastic benchmark, the bounds from the law of the iterated logarithm, and the analytical solutions for the distribution of times to return to the origin (Fig. 6a and b), with $R^2 = 0.9953$ (Table 1). The analytical exact and approximate solutions of the stochastic Poisson model (Eqs. 34 and 35) predict a monotonically declining probability density with increasing return times (Fig. 6b). The analytical approximation of the age distribution (Eq. 36, Fig. 6b) underpredicts the ages modelled by the stochastic benchmark for small ages in comparison to the exact solution but provides an exact description of the right-hand power-law tail (Fig. 6b).

The scaling predicted in the analytical equations also agrees well with the selected field and experimental datasets. First, the $-3/2$ power-law scaling (Eq. 35) for the distribution of times to return to the origin is consistent with the data from the Little Missouri River at Watford, North Dakota, USA (Everitt, 1968); the South River near Waynesboro, Virginia, USA (Skalak and Pizzuto, 2010); and the Powder River between Moorhead and Broadus, Montana, USA (Huffman et al., 2022) (Fig. 7; $R^2 = 0.8434$, 0.8168, and 0.5576, respectively). Second, in the evolution of the channel belts in analogue experiments, we can clearly identify a drift phase (Fig. 8). This phase is apparent as a square root scaling of channel-belt width as a function of time (Eq. 25). We find $q_L/k = 2.15 \times 10^{-5}\,\mathrm{m^2\,s^{-1}}$ for Run 5 ($R^2 = 0.9995$) and $q_L/k = 2.62 \times 10^{-5}\,\mathrm{m^2\,s^{-1}}$ for Run 7 ($R^2 = 0.9960$). The exponential phase (Eq. 12) can also be fitted independently (see Bufe et al., 2019). However, the data resolution is not good enough to fit both relationships with consistent parameter values. Essentially, the resulting unconfined channel-belt width $W_0$ depends on the subjective choice of which data points to include into the fit.

## 5   Discussion

### 5.1   Model predictions and overview

Using the Poisson concept for the formation and evolution of channel belts, we derived a range of results that hold implications for fluvial geomorphology, quantitative landscape evolution studies, and river management (Table 2). The stochastic treatment allowed us to theoretically quantify one of the two unconstrained parameters in the model of Turowski et al. (2024). As such, apart from the factor of proportionality $k$ in the definition of the switching timescale $\lambda$ (Eq. 2), all of the model parameters can be directly related to channel geometry and hydraulics. In particular, to parameterize the model, one needs measurements of flow depth $h$, channel width $W_C$, and the lateral transport capacity $q_L$. The former two have been routinely measured in the field. Yet, natural river discharge changes over time, and it is currently unclear which flood size is responsible for setting the channel belt in the long-term channel dynamics. The lateral transport capacity depends on water discharge, sediment supply, and granulometry of a particular river (Bufe et al., 2019). The precise relationship is debated (e.g. Bufe et al., 2019; Constantine et al., 2014; Ielpi and Lapôtre, 2019; Wickert et al., 2013) and likely depends on the characteristics of the particular river, for example its planform type (Greenberg et al., 2024; Nyberg et al., 2023).

Our model has been constructed assuming a single laterally migrating channel as it constructs a channel belt between two avulsion events (Bridge and Leeder, 1979; Nyberg et al., 2023). Yet, many rivers are braided or anastomosing, featuring multiple channels. It is not clear at the moment whether our model can also be applied to those rivers. A number of points can be made, though, based on generic arguments and observations (Turowski et al., 2024). First, multiple channels would add a complexity to the model that is beyond the first-

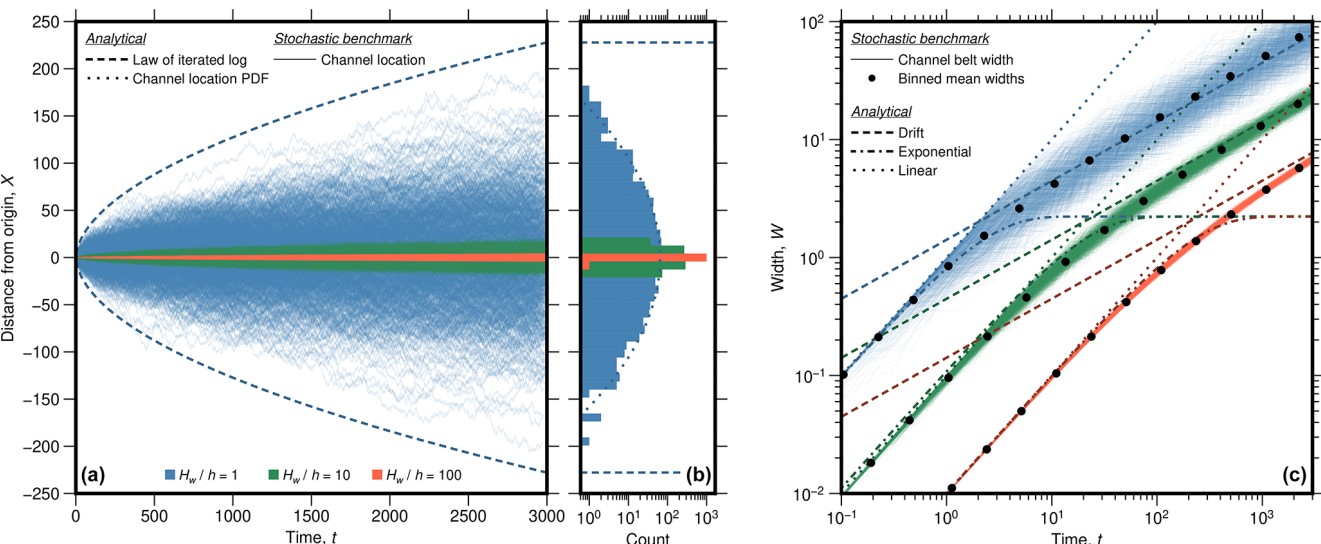

**Figure 2.** Temporal evolution of channel-belt width in the stochastic benchmark and comparison between the stochastic benchmark and the analytical solutions. **(a)** Modelled migration paths through time (solid coloured lines), bounded by the law of the iterated logarithm (dashed line, Eq. 32), i.e. the area that the river almost surely does not cross. Similar plots with longer runtimes can be found in Figs. 5a and 6a. The blue lines show the evolution of an unconfined river ($H_W/h = 1$), the green lines show a moderately confined case ($H_W/h = 10$), and the orange lines show a highly confined case ($H_W/h = 100$). **(b)** Location density at $t = 3000$. The dotted line gives the theoretically expected normal distribution for the unconfined case (blue), and the dashed line marks the law of the iterated logarithm. **(c)** Average width evolution with time, showing the analytical expressions for the linear (dotted lines, Eq. 1), exponential (dash-dotted lines, Eq. 12), and drift phases (dashed lines, Eq. 25). Fine solid lines show the outputs from the numerical simulation, and black circles show the mean widths of these simulations in bins spaced logarithmically in time. Standard errors of the means are smaller than the symbols. $R^2$ values for the comparisons are given in Table 1.

**Table 1.** Statistics for the comparison of the analytical results with the stochastic benchmark and the data.

| Test | Equation no. | Figure no. | $R^2$ |
|---|---|---|---|
| Comparison of analytical equations to the stochastic benchmark | | | |
| Normal distribution of channel positions | | 2b | 0.9550 |
| Width increase in the exponential phase | 12 | 2c | 0.9995 |
| Width increase in the drift phase | 25 | 2c | 0.9966 |
| Drift velocity as a function of width | 29 | 3b | 0.9999 |
| Drift velocity as a function of lateral transport capacity | 29 | 3c | 0.9999 |
| Verification of the value of $c$ | 30 | 4 | 0.9997 |
| First passage distribution | 33 | 5b | 0.9991 |
| Return time distribution, exact solution | 34 | 6b | 0.9953 |
| Return time distribution, right-hand tail | 35 | 6b | 0.9995 |
| Return time distribution, approximate solution | 36 | 6b | 0.9980 |
| Comparison of analytical equations to data | | | |
| Return time distribution, fit to Everitt (1968) | 35 | 7 | 0.8434 |
| Return time distribution, fit to Skalak and Pizzuto (2010) | 35 | 7 | 0.8168 |
| Return time distribution, fit to Huffman et al. (2022) | 35 | 7 | 0.5576 |
| Drift in the experiment in Run 5 | 25 | 8a | 0.9995 |
| Drift in the experiment in Run 7 | 25 | 8b | 0.9960 |

**Table 2.** Overview of the analytical equations.

| Result | Comment | Equation no. | Equation |
|---|---|---|---|
| Channel lateral migration speed | Suggested by Bufe et al. (2019) from experimental data. | 1 | $V = \dfrac{q_L}{H_+}$ |
| Average switching rate | Derived by Turowski et al. (2024). | 2 | $\lambda = k\dfrac{q_L}{h^2}$ |
| Unconfined steady-state channel-belt width | Derived by Turowski et al. (2024). | 4 | $W_0 = \dfrac{c}{k}h + W_C$ |
| Steady-state valley width | Includes uplift and lateral sediment supply as additional input parameters in comparison to Eq. (4). Derived by Turowski et al. (2024). | 5 | $W_V = \left(\dfrac{q_L - q_H}{U}\right)\ln\left\{1 + \dfrac{U(W_0 - W_C)}{q_L}\right\} + W_C$ |
| Exponential approach to steady state | Evolution equation in the exponential phase. | 12 | $W(t) = W_0 - (W_0 - W_C)\exp\left\{-\dfrac{t}{\tau}\right\} + W_C$ |
| Governing timescale, unconfined case | To be used in Eq. (12). | 13 and 14 | $\tau = \dfrac{(W_0 - W_C)(H_W - h)}{q_L}$   $\dfrac{H_+}{q_L} = \left(\dfrac{H_W}{h} - 1\right)\dfrac{c}{\lambda}$ |
| Governing timescale, confined case | To be used in Eq. (12). | 18 | $\tau = (W_0 - W_C)\dfrac{h}{\lambda}$ |
| Square root widening | Average increase in area affected by the channel in the drift phase, after the steady state width has been reached. | 25 | $W_{\mathrm{Drift}}(t) = \sqrt{\dfrac{2}{k}\dfrac{h}{H_W}q_L t} + W_C$ |
| Average drift speed | Average drift speed in the drift phase, assuming the channel belt keeps a constant width. | 29 | $V_{\mathrm{Drift}} = \dfrac{1}{\sqrt{2k}}\sqrt{\dfrac{h}{H_W}(W_0 - W_C)}$ |
| Channel-belt limits | Law of the iterated logarithm as an envelope to the area that the channel is unlikely to leave. Only valid for unconfined channel belts. | 32 | $X(t) = \pm 2\dfrac{h}{k}\sqrt{2\dfrac{\lambda t}{k}\ln\left\{\ln\left\{2\dfrac{\lambda t}{k}\right\}\right\}}$ |
| First passage time distribution | Distribution of times needed to reach a point that is a distance $b$ from the origin (Lévy distribution). | 33 | $PDF_{\mathrm{FP,R}}(T_{\mathrm{FP}}) = \dfrac{\lvert b\rvert}{\sqrt{2\pi\frac{h}{H_W}\frac{q_L}{k}T_{\mathrm{FP}}^3}}\exp\left\{\dfrac{-b^2}{2\frac{h}{H_W}\frac{q_L}{k}T_{\mathrm{FP}}}\right\}$ |
| Distribution of times needed to return to the origin | This is equivalent to the sediment residence time distribution, or the age distribution of sediments, assuming a single deposition and remobilization event. The integral equation does not have an analytical solution. | 34 | $PDF_{\mathrm{RT}}(T_R) = \displaystyle\int_0^{\frac{h}{H_W}Vt} \dfrac{\frac{h}{H_W}V}{\sqrt{2\pi\frac{h}{H_W}\frac{\lambda}{k}}}\exp\left\{\dfrac{-\lambda}{V}\Delta x\right\} \dfrac{\lvert\Delta x\rvert}{\sqrt{2\pi\frac{h}{H_W}\frac{q_L}{k}\left(T_R - \frac{\Delta x}{\frac{h}{H_W}V}\right)^3}}\exp\left\{\dfrac{-\Delta x^2}{2\frac{h}{H_W}\frac{q_L}{k}\left(T_R - \frac{\Delta x}{\frac{h}{H_W}V}\right)}\right\}\,d\Delta x$ |
| Analytical right-hand tail of the distribution of times needed to return to the origin | An analytical solution for the right-hand tail of Eq. (34). | 35 | $PDF_{\mathrm{RT}}(T_R \gg 0) = \dfrac{1}{\sqrt{2\pi}}\left(\dfrac{h}{H_W}\lambda T_R\right)^{-3/2}$ |
| Analytical approximation for the distribution of times needed to return to the origin | Analytical approximation for Eq. (34). | 36 | $PDF_{\mathrm{RT}}(T_R) \approx \dfrac{1}{\sqrt{2\pi}}\dfrac{\frac{h}{H_W}}{1 + a\left(\frac{h}{H_W}\lambda T_R\right)^{3/2}}, \; a = \left(\dfrac{5}{2}\right)^3\left(\dfrac{3}{2\pi}\right)^{3/2} = 1.1135$ |
| Value for time scaling constant $c$ | The constant relates the average switching rate $\lambda$ to the effective switching time $\Delta T$ (see Eq. 3). | 30 | $c = \dfrac{1}{\sqrt{2}\,V_{\mathrm{Drift}}} = 2.2285$ |

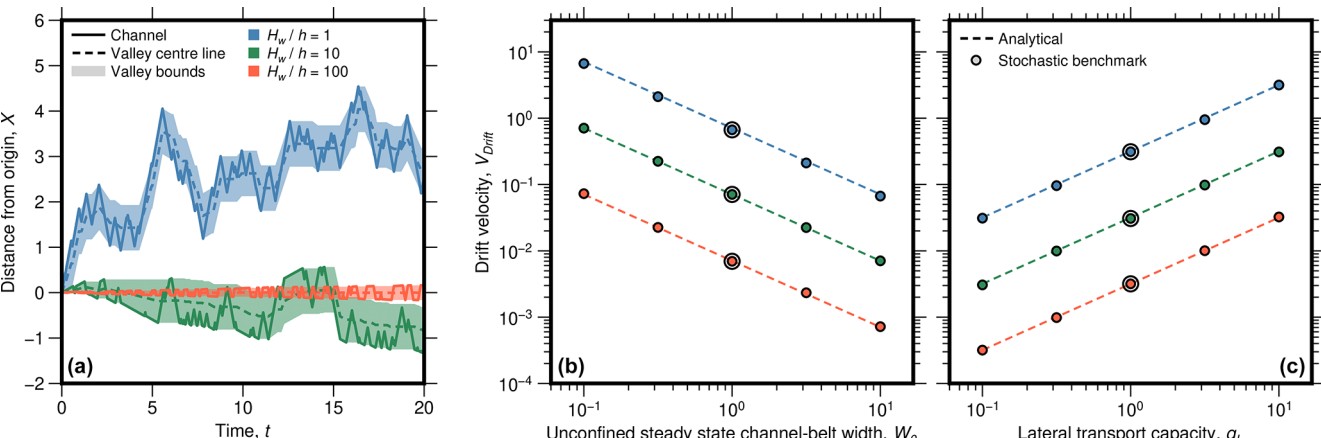

**Figure 3.** Lateral drift speed of channel belts at a constant steady-state width for the drift phase. For the calculation, the channel-belt width was fixed to the steady-state width; i.e. whenever the channel widened the channel belt on one side, the width was reduced by the same amount on the other side. **(a)** Channel location as a function of time for different degrees of confinement (same colour code as in Fig. 2). Note that **(a)** does not show the entire calculated trajectories; average drift velocities were measured after 10 000 steps. **(b)** Average drift speeds as a function of steady width from the stochastic benchmark are shown as circles. The analytical predictions (dotted lines) of Eq. (29) fit the numerical results well. **(c)** Average drift speed as a function of lateral transport capacity with the same symbology as in **(b)**. Larger circles in **(b)** and **(c)** show simulations plotted in **(a)**. $R^2$ values for the comparisons are given in Table 1.

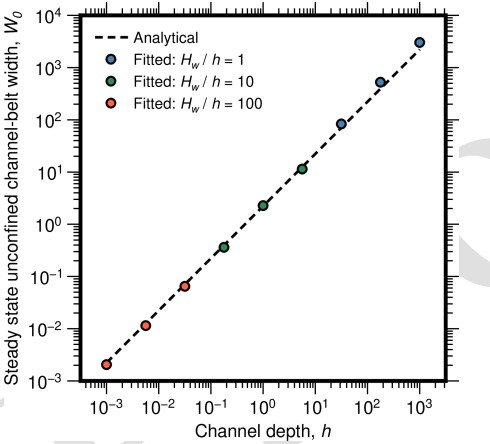

**Figure 4.** Verifying the value of the constant $c$ (see Eq. 30) by comparing the unconfined steady-state channel-belt width obtained from fits to the stochastic Poisson model (Fig. 1) to channel depth for varying simulations. We set channel width to $W_C = 0$ and $k = 1$ for these simulations. Then, the steady-state channel-belt width and flow depth should be proportional with a constant of proportionality equal to $1/c$ (Eq. 4). The dashed line gives the theoretically expected relationship with $c = 2.2285$ (Eq. 30). The results show that the value of $c$ is the same for unconfined and confined channel belts. $R^2$ values for the comparisons are given in Table 1.

order treatment developed here. Second, Dong and Goudge (2022) argued that the belt width of both single-thread and braided channels follow a systematic trend. This may indicate that the generic model equations can be extended to encompass the belt width of braided rivers. Third, the channels in the Bufe et al. (2016) experiments frequently split

into multiple channels. Nevertheless, the square root scaling expected for the drift phase can be observed (Fig. 8), and observed narrowing of valleys in response to uplift closely follows the predicted relationship (Eq. 5) (see Turowski et al., 2024). These results may indicate that multiple channels lead to an average rate and pattern of lateral migration similar to those of a single migrating channel. Fourth, Bufe et al. (2019) found that $q_L$ scales approximately linearly with water discharge in experiments featuring multiple channels. This indicates that the area affected by migrating channels is independent of the detailed distribution of water between single or multiple channels. How different channels interact by merging, splitting, and crossing, and how this affects their lateral migration speed and dynamics needs to be investigated in future work.

## 5.2 The effect of uplift

In our model, we have not explicitly considered the role of uplift or net incision in the channel-belt width. Uplift increases the bank height encountered by the channel in lateral motion (Eq. 1) and thereby slows it down. Turowski et al. (2024) included uplift in their steady-state valley width model and demonstrated that a competition between uplift and lateral mobility of the channel, described by the lateral transport capacity, determines the final width of the valley. Yet, the inclusion of uplift in the stochastic Poisson model developed herein would introduce considerable complexity into the equations. It seems unlikely that analytical solutions are possible. Here, we suggest a simple approach to circumvent this problem. We use Eqs. (1) to (5) to define an effective

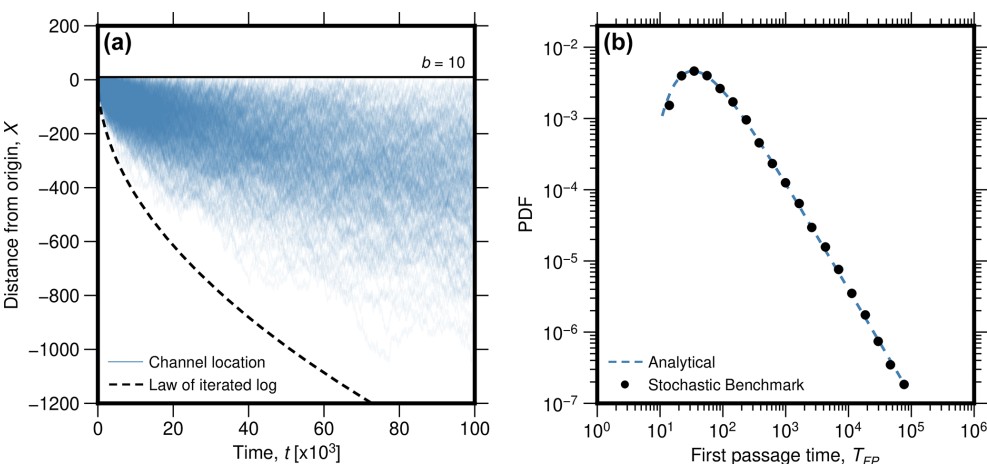

**Figure 5.** The results for the first passage distribution. **(a)** Paths of models to investigate the time distribution to reach a point that is a distance $b$ from the origin (horizontal black line). The dashed line gives the expectation from the law of the iterated logarithm (Eq. 32). In comparison to Fig. 2a, substantially longer runs in time are shown here. **(b)** First passage time distribution of the stochastic benchmark (black dots show binned means) in comparison to the analytical solution (dotted blue line, Eq. 33). $R^2$ values for the comparisons are given in Table 1.

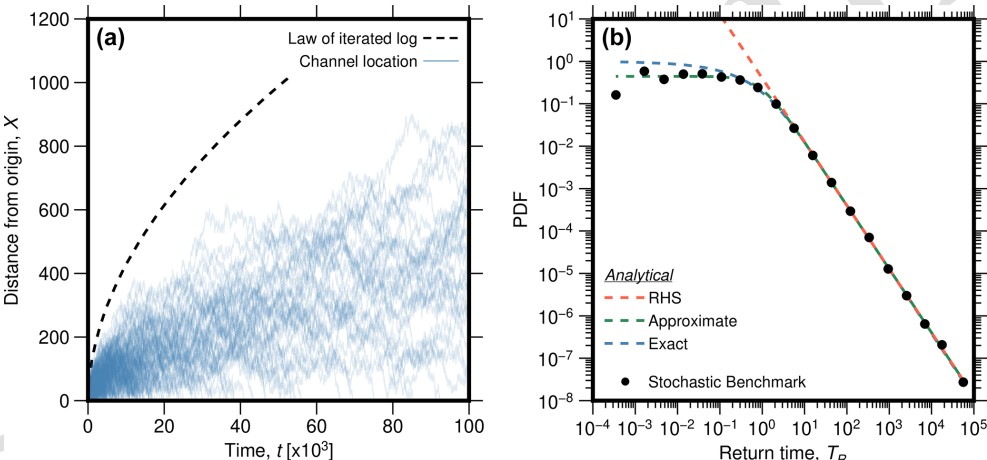

**Figure 6.** The analytical results for the return time distribution, equivalent to the age distribution of sediments stored in the channel belt, and comparison to data. **(a)** Paths of 10 000 models to investigate the time distribution for the return to the origin. Once a model path reached the origin, later time steps are not plotted. The dashed line gives the prediction of the return time from the law of the iterated logarithm (Eq. 32). In comparison to Fig. 2a, substantially longer runs in time are shown here. **(b)** Modelled return time distribution (black dots show binned means) compared to the exact analytical solution (blue, Eq. 34) and the power-law decay in the right-hand-side (RHS) tail with an exponent of $-3/2$ (red, Eq. 35). The analytical approximation (green, Eq. 36) is also shown. $R^2$ values for the comparisons are given in Table 1.

lateral migration speed $\overline{V_U}$ [L T$^{-1}$] in an uplifted area:

$$W = \frac{c\overline{V_U}}{\lambda} + W_C$$
$$= \frac{q_L}{U} \ln\left\{ 1 + \frac{U(W_0 - W_C)}{q_L} \right\} + W_C. \tag{37}$$

Solving for $\overline{V_U}$, this yields

$$\overline{V_U} = \frac{k}{c}\frac{V^2}{U} \ln\left\{ 1 + \frac{U(W_0 - W_C)}{q_L} \right\}. \tag{38}$$

We thus obtain an effective variance:

$$\begin{aligned}
\mathrm{VAR} =& \frac{2}{k}\frac{h}{H_W}\frac{\overline{V_U}^2}{\lambda} t \\
=& \frac{2}{k}\left(\frac{k}{c}\right)^2 \frac{h}{H_W}\frac{V^4}{U^2}\frac{t}{\lambda}\ln^2\left\{ 1 + \frac{U(W_0 - W_C)}{q_L} \right\} \\
& \times 2\left(\frac{k}{c}\right)^2 \frac{h}{H_W}\frac{q_L^2}{(W_0 - W_C)U^2}q_L t \\
& \times \ln^2\left\{ 1 + \frac{U(W_0 - W_C)}{q_L} \right\}.
\end{aligned} \tag{39}$$

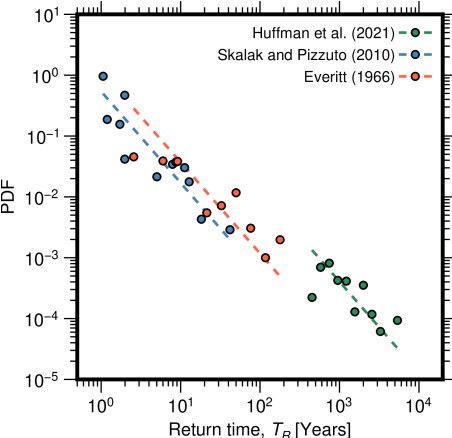

**Figure 7.** Floodplain age data from Everitt (1968), Skalak and Piz­zuto (2010), and Huffman et al. (2022) are consistent with the $-3/2$ power-law tail (Eq. 35). $R^2$ values for the fits are given in Table 1.

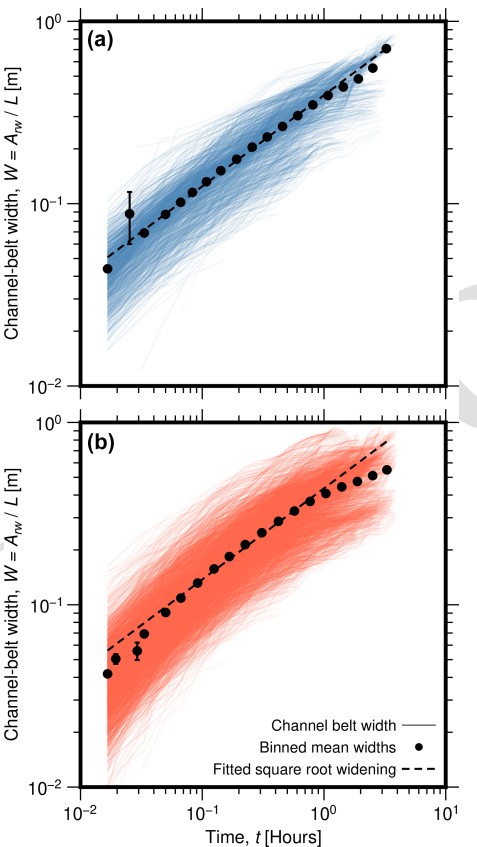

**Figure 8.** Temporal evolution of the cumulative inundated area in the experiments of Bufe et al. (2016b, 2019), with data from **(a)** Run 5 (blue) and **(b)** Run 7 (red). Black dots give binned means, and error bars show the standard errors of the means (mostly smaller than the symbols). The dashed line is the fitted square root widen­ing relationship with time that can be expected for the drift phase (Eq. 25). $R^2$ values for the fits are given in Table 1.

Equation (39) can be used in Eq. (19) for the drift to ac­count for uplift. Other results also have to be updated accord­ingly. The approach outlined above needs to be benchmarked with numerical simulations or field or experimental data.

## 5.3 First passage and floodplain age distributions

The Lévy distribution (Eq. 33) describes the time needed until the channel moves a particular distance away from its starting location. When integrated to infinity, the distribution has an infinite mean and variance. Nevertheless, it could be used, for example, for assessing the risk of the destruction of a building near a river channel within a given time span.

Lateral river dynamics determine the reworking of sedi­ment in the floodplain and, therefore, determine storage times and sediment ages (e.g. Bradley and Tucker, 2013). This has, for example, implications for chemical alteration of flood­plain sediments, such as chemical weathering and organic carbon oxidation (e.g. Scheingross et al., 2021; Repasch et al., 2020; Torres et al., 2017). It has frequently been found that residence time distributions are highly skewed and that the mean residence time of sediment is much larger than their median residence time (e.g. Carretier et al., 2020; Pizzuto et al., 2017). Measurements of the distribution of floodplain ages have yielded a variety of contrasting behaviour (Piz­zuto et al., 2017). The right-hand tail of the distribution of field data has been characterized by both an exponential (e.g. Huffman et al., 2022; Lancaster and Casebeer, 2007) and a power-law function (e.g. Bradley and Tucker, 2013; Pizzuto et al., 2017), in the latter case with exponents ranging from about $-0.7$ to about $-1.5$ (e.g. Lancaster et al., 2010; Pizzuto et al., 2017; Skalak and Pizzuto, 2010). Pizzuto et al. (2017) used a random walk to model the stochastic downstream mo­tion of sediment to predict power-law travel-time distribu­tions with exponents that decrease with increasing length of the river system.

Bradley and Tucker (2013) suggested that the Lévy distri­bution is suitable to model the distribution of floodplain ages. Analogous to our result for the age distribution (Eq. 34), the Lévy distribution features a power-law right-hand tail with a scaling exponent of $-3/2$ (Eq. 33). However, it strongly underpredicted the likelihood of small ages as generated by the Bradley and Tucker (2013) numerical model. The Lévy distribution has been derived for the time of the first passage of a point a pre-selected distance from the origin (Eq. 33), and this distance cannot be equal to zero in the assumptions of the derivation. It is therefore not the correct distribution for the times to return to the origin. We derived a probabil­ity distribution for the time to return to the origin (Eq. 34). The right-hand tail of the residence time distribution (Eq. 35) exhibits the same scaling of the right-hand tail of the Lévy distribution (Eq. 33), a power law with an exponent of $-3/2$ (Fig. 6b). In fact, this scaling is valid for any symmetric ran­dom walk and should be independent of the precise assump­tions used to set up such a random walk. It implies that the

return time distribution has both an infinite mean and standard deviation when integrated to infinity, similar to the distribution of first passage. This result implies that the mean age measured for a sediment body within a channel belt does not converge to a fixed value but depends on the time since the onset of fluvial activity, no matter how long ago this onset occurred. The result implies that statements on the age of sediment in floodplains, or their chemical alteration, always have to be made with respect to the total age of the floodplain. A long-term average at steady state is never achieved. Further, it implies that some fluvial deposits are likely to survive for long times, storing information about the floodplain evolution and the history of river systems (cf. Carretier et al., 2020). The increase in the mean sediment residence time $\overline{T_R}$ can be obtained by integrating the age distribution (Eq. 34) multiplied with time, as in the integration for the mean. We can obtain the limit behaviour for old river systems by integrating over Eq. (35):

$$
\begin{aligned}
\overline{T_R}(t) &= \int_0^{T_A} \frac{\lambda}{\sqrt{2\pi}} \left( \frac{h}{H_W} \lambda t \right)^{-3/2} t \, \mathrm{d}t \\
&= \sqrt{\frac{2}{\pi} \left( \frac{H_W}{h} \right)^3 \frac{T_A}{\lambda}}.
\end{aligned}
\tag{40}
$$

Here, $T_A$ is the time since the formation of the channel belt. The mean residence time thus increases with the square root of time in this limit. In combination with Eq. (35), Eq. (40) can be used to estimate the age of a channel belt from sediment age data.

Our prediction of the $-3/2$ scaling exponent in the age distribution (Eqs. 34 and 35) does align with some, but not all, of the measurements reported in the literature (see Pizzuto et al., 2017). It is consistent with the data of Everitt (1966), Skalak and Pizzuto (2010), and Huffman et al. (2022) that we digitized for the present study (Fig. 7) but not with the datasets reported, for example, by Lancaster et al. (2010). For our comparison, we selected datasets that, on first glance, comply with the assumptions underlying our stochastic Poisson model. The model framework is strictly valid only for processes that can be modelled by a lateral random walk of a single channel in an infinite domain. As such, we expect it to apply to single-thread channels without major tributaries that are undisturbed by processes other than fluvial erosion and deposition. Further, the $-3/2$ scaling applies to channels that are short enough such that sediment, once it is eroded, is not redeposited within the system but evacuated downstream. Alternatively, the scaling could apply to data measured with dating methods where the date is reset after remobilization of sediment, for example optically stimulated luminescence (e.g. Madsen and Murray, 2009). Multiple episodes of deposition and erosion within the same system yield a power-law tail with an exponent that depends on the system size (Pizzuto et al., 2017). This exponent should, generally, be smaller than $-3/2$ because redeposition will increase the relative fraction of old sediment. Even in short systems, the derived age distribution (Eq. 34) cannot be expected to be universally applicable. We expect that channels confined in a narrow valley, or those in which processes other than lateral channel migration can deposit, evacuate, or mobilize sediment, show different scaling behaviour. For example, the channels studied by Lancaster and Casebeer (2007) and Lancaster et al. (2010) are located in confined valleys where debris flows regularly supply and mobilize sediment and exhibit age distributions with power-law scaling exponents of the order of $-0.7$. In narrow confined settings, sediment deposition and erosion may not be adequately described by a random walk. Further, the disturbance of fluvial deposits and lateral sediment supply by debris flows or hillslope processes may have a large effect on the age distribution.

## 5.4   Parameter estimation and further tests

The model contains a single dimensionless scaling factor, $k$, which is the factor of proportionality of the rate of switches of direction of motion of the channel $\lambda$ and the ratio of the lateral transport capacity $q_L$ to the square of the flow depth $h$ (Eq. 2). This parameter sets the unconfined channel-belt width (Eq. 4). Two strategies for measuring this parameter appear from our results. First, exploiting Eq. (2) relies on direct measurements of the switching rate, as well as flow depth and $q_L$. The switching rate $\lambda$ can also be measured from the age distribution of sediment (Eq. 41). Second, the width of the channel belt can be related to flow depth and channel width using Eq. (4). Both approaches seem more promising in an experimental setting than in nature because the necessary parameters can either be controlled or measured directly. In the field, it may be possible to obtain suitable data, for example, from time series of orthophotos of river reaches (e.g. Nyberg et al., 2023; Greenberg and Ganti, 2024; Greenberg et al., 2024) in combination with gauging data. Testing for the consistency of both approaches would be a strong method to falsify or validate the model.

Our model is constructed at the reach scale of the channel and does not include detailed descriptions of fluvial processes. Yet, it should be possible to relate it to process-based models. Here, we make a tentative relation to models of meandering channels, which are available at different degrees of complexity (e.g. Edwards and Smith, 2002; Ikeda et al., 1981). Camporeale et al. (2005) studied models of meandering rivers at increasing levels of hydraulic detail. They concluded that the steady-state statistics of the meander belt are determined by only two parameters, regardless of the complexity of the model. These are a length scale $D_0$ [L] proportional to the ratio of flow depth and the friction coefficient for open channel flow $C_f$,

$$
D_0 = \frac{h}{2C_f},
\tag{41}
$$

and a timescale $T_0$ [T], given by

$$T_0 = \frac{D_0^2}{W_C U_f E} . \quad (42)$$

Here, $U_f$ [LT$^{-1}$] is the mean streamwise flow speed and $E$ [–] a dimensionless bank erodibility coefficient. Using their model considerations together with field observation, Camporeale et al. (2005) found that the meander belt width $W_{MB}$ can be described by

$$W_{MB} = \alpha D_0 = \frac{\alpha h}{2 C_f} . \quad (43)$$

Here, $\alpha$ [–] is a dimensionless proportionality coefficient with a value of 40 to 50. We can use Eqs. (41)–(43) to make a tentative connection between our landscape-scale random walk model and the reach-scale meandering models. First, we note that both models suggest that channel-belt width is proportional to flow depth (see Eq. 4). Comparing Eqs. (4) and (43), we suggest that $k_0$ scales as

$$k_0 = \frac{c}{k} = \frac{\alpha}{2 C_f} . \quad (44)$$

As such, we expect $k$ to scale with the friction coefficient. Assuming $C_f = 0.05$ and $\alpha = 50$ (see Camporeale et al., 2005), we obtain $k = 0.0045$ and $k_0 = 500$. Second, we can assume that the governing timescale $\tau$ (Eqs. 13 and 14) is proportional to $T_0$. Equating Eqs. (14) and (42), and substituting Eqs. (2), (41), and (43), we obtain

$$\begin{aligned}
\frac{c}{\lambda} &= \frac{ch^2}{kq_L} = \frac{\alpha}{2 C_f} \frac{h^2}{q_L} = \frac{D_0^2}{W_C U_f E} \\
&= \left( \frac{h}{2 C_f} \right)^2 \frac{1}{W_C U_f E} .
\end{aligned} \quad (45)$$

Equation (45) can be solved for $q_L$ to give

$$q_L = 2 \alpha C_f W_C U_f E . \quad (46)$$

We can obtain some of the parameter values from the data used in this study. From fits to the floodplain age distributions, we obtain $\lambda = 0.12\,\mathrm{yr}^{-1}$ (Everitt, 1966), $\lambda = 0.55\,\mathrm{yr}^{-1}$ (Skalak and Pizzuto, 2010), and $\lambda = 0.00097\,\mathrm{yr}^{-1}$ (Huffman et al., 2022). Note that we assumed an unconfined channel belt for determining $\lambda$; i.e. we set $H_W = h$. In the case of confinement, the estimates change with the ratio of the flow depth and the height of the confining walls (Eq. 35). The numbers for the mean rate of switching seem plausible, varying from biannual switches (Skalak and Pizzuto, 2010) to once in a thousand years (Huffmann et al., 2022). The estimates should be further refined with detailed case studies.

## 5.5 Beyond the evolution of single cross sections

In the stochastic Poisson model developed herein, we concentrated on a single cross section, making the assumption that each cross section evolves independently of those upstream and downstream. This assumption is likely to be a simplification when applied to real river systems. In particular, we can expect that a channel that locally moves laterally far from the channel position upstream and downstream is pulled back towards the centre. That is, a channel within a particular cross section of the valley is less likely to further migrate laterally in the same direction if within the cross sections upstream and downstream into which the channel has not migrated as far or is moving in the opposite direction. This effect can be included in the model by modulating the probability of switching direction $\lambda$ within the cross section of interest depending on the position of its channel with respect to the entire river system or to the cross sections immediately upstream and downstream. We suggest that the behaviour can be modelled by an Ornstein–Uhlenbeck process (e.g. Uhlenbeck and Ornstein, 1930), similar to the Langevin equation (Langevin, 1908), which includes a term that increases the probability to move back towards the origin as a function of the distance from it. It is beyond the scope of the present contribution to develop such a model. We expect that the suggested approach will yield a Gaussian distribution of channel positions, with similar results to those derived herein, but with additional dimensionless scaling factors in the variances.

## 6 Conclusion

We have described the temporal evolution of unconfined and confined channel-belt width in the framework of a random walk. The temporal evolution can be described in three phases, which are associated with distinct timescales. First, channel belts grow linearly before the channel switches direction. Then, the channel-belt width increases exponentially until the steady-state width is achieved. Finally, the channel belt enters the drift phase, where it grows on average with the square root of time. Using the mathematics of random walks, we derived a range of other results, including the limits of the channel belt (law of the iterated logarithm), the distribution of times to arrive at a particular distance from the origin (first passage distribution), and the distribution of times until the channel returns to its origin, which is equivalent to the distribution of sediment ages within the channel belt. All results directly connect to hydraulic parameters such as flow depth, channel width, and the lateral transport capacity, and the model contains a single free parameter that needs to be calibrated on data. To validate the stochastic Poisson model, model predictions were compared to numerical simulations of channel-belt evolution, field data of floodplain ages, and analogue experiments. The comparisons strongly support the basic assumption that channel-belt development can be described by a random walk. The predicted scaling exponent for the age distribution of floodplain sediments is consistent with observations from streams that were selected to

closely align with the assumption made in the model. In experimental data (Bufe et al., 2016a, b, 2019), average widening proceeds with the square root of time, as expected for the drift phase. Recent global datasets on channel belts derived by automatic processing of remote sensing data (e.g. Dong and Goudge, 2022; Greenberg and Ganti, 2024, Greenberg et al., 2024; Nyberg et al., 2023) provide opportunities for comprehensive testing of the model. We have provided a range of analytical results (Table 2) that allow easy comparison of theory and data. These can also be directly implemented into landscape evolution models without major numerical costs, allowing a more comprehensive and realistic depiction of landscape dynamics. The stochastic Poisson model can in principle be used for forward predictions in the context of river management, flood hazard mitigation, and stream restoration. In addition, our work provides a theoretical framework to interpret observational data related to fluvial landscape evolution, nutrient cycling, and inverting fluvial strata for paleo-hydraulic conditions. In summary, all parameters of the stochastic Poisson model have a direct physical interpretation, and there is a single free, dimensionless scaling parameter that needs to be informed by data. As such, our approach can bridge across spatio-temporal scales and connect landscape-scale models with those operating on the process scale.

## Appendix A

**Table A1.** Symbols and notation.

| Symbol | Parameter | First appears in Eq. |
| --- | --- | --- |
| $\alpha$ | Dimensionless proportionality coefficient with a value of 40 to 50 [–] | 42 |
| $\lambda$ | Rate parameter of the Poisson process describing the switch in the direction of river motion [$T^{-1}$] | 2 |
| $\tau$ | Governing timescale for the transient approach to a steady state [T] | 12 |
| $a$ | Dimensionless constant approximately equal to 1.1135 [–] | 36 |
| $b$ | Distance of an point of interest from the river channel at $t = 0$ [L] | 33 |
| $c$ | Dimensionless constant approximately equal to 2.2285 [–] | 3 |
| $C_f$ | Open-channel-flow friction coefficient [–] | 40 |
| $D_0$ | Characteristic length scale of meander belts [L] | 40 |
| $E$ | Dimensionless bank erodibility coefficient [–] | 41 |
| $h$ | Flow depth [L] | 2 |
| $H_+$ | Height of the riverbank in the direction of river motion [L] | 1 |
| $H_W$ | Height of the walls confining the channel belt [L] | 17 |
| $k$ | Dimensionless constant of order $10^{-2}$ to $10^{-3}$ [–] | 2 |
| $k_0$ | Dimensionless constant of order $10^2$, defined by $c/k$ [–] | 4 |
| $n$ | Number of stochastic events, generally used for the number of steps in the random walk [–] | 6 |

| Symbol | Parameter | First appears in Eq. |
| --- | --- | --- |
| $m$ | Number of pairs of steps in the random walk, generally defined as $n/2$ [–] | |
| $q_H$ | Rate of lateral sediment supply from hillslopes or valley walls per channel length [$L^2 T^{-1}$] | 5 |
| $q_L$ | Lateral transport capacity, i.e. the amount of sediment that the channel can move by lateral erosion per unit channel length per unit time [$L^2 T^{-1}$] | 1 |
| $P$ | Fraction of time that a river spends at any of its channel belt margins [–] | 9 |
| $P_{confined}$ | Fraction of time that a river spends at any of its channel belt margins for a confined belt [–] | 15 |
| $S$ | Dimensionless envelope distance for the channel belt in the law of the iterated logarithm [–] | 31 |
| $t$ | Time [T] | 4 |
| $t^*$ | Dimensionless time [–] | 31 |
| $\Delta t$ | Average switching timescale in the Poisson process [T] | 6 |
| $T_0$ | Characteristic timescale of meander belts [T] | 41 |
| $\Delta T$ | The characteristic length of time the river moves on average in the same direction [T] | 3 |
| $T_A$ | Time since the formation of the channel belt; age of the channel belt [T] | 40 |
| $T_{FP}$ | First passage time, first point in time when the channel reaches a point of interest located a distance $b$ from the channel at $t = 0$ [T] | 33 |
| $T_R$ | Time needed to return to the origin for the first time [T] | 34 |
| $\overline{T_R}$ | Mean residence time of sediment [T] | |
| $T_{SS}$ | Time at which the steady-state width is reached [T] | 27 |
| $T_W$ | Waiting times between events in a Poisson process [T] | 7 |
| $U$ | Uplift rate [$L T^{-1}$] | 5 |
| $U_f$ | Mean streamwise flow speed [$L T^{-1}$] | 41 |
| $v$ | Lateral speed of the river as it reaches valley floor margins, i.e. wall toes [$L T^{-1}$] | 15 |
| $V$ | Lateral migration speed, i.e. the speed of river migrating back and forth across the valley floor [$L T^{-1}$] | 1 |
| $\overline{V}$ | Average lateral channel migration speed in a confined channel belt [$L T^{-1}$] | 23 |
| $V_{Drift}$ | Average lateral speed of a channel belt with constant width during the drift phase [$L T^{-1}$] | 29 |
| $VAR_{CCB}$ | Variance of a confined channel-belt width [$L^2$] | 24 |
| $VAR_{UCB}$ | Variance of an unconfined channel-belt width [$L^2$] | 19 |

| Symbol | Parameter | First appears in Eq. |
|---|---|---|
| $W$ | Channel-belt width [L] | 5 |
| $W_c$ | River channel width [L] | 4 |
| $W_{Drift}$ | Width of channel belt in the drift phase [L] | 19 |
| $W_{MB}$ | Width of a meander belt [L] | 42 |
| $W_V$ | Valley floor width [L] | 5 |
| $W_0$ | Unconfined channel-belt width [L] | 4 |
| $\Delta x$ | Distance travelled by the channel before switching direction for the first time [L] | 34 |
| $X$ | Envelope distance for the channel belt in the law of the iterated logarithm, dimensional version of $S$ [L] | 32 |
| $X_{Drift}$ | Average distance drifted in the drift phase [L] | 26 |

**Data availability.** Raw data for the experimental datasets are stored on the SEAD repository of Bufe et al. (2016a) with the identifier https://doi.org/10.5967/M0CF9N3H. Derived quantities have been compiled from Bufe et al. (2016a, b) and Bufe et al. (2019). Sediment age data were digitized from the respective publications. Scripts used to generate Figs. 2–7 are available in the publication by McNab (2024) with identifier https://doi.org/10.5281/zenodo.12806574.

**Author contributions.** JMT, AB, and ST conceived and conceptualized this study. JMT designed and developed the theoretical approach, derived the equations with input of FM and AB, and wrote the paper. FM wrote the scripts for the stochastic benchmark and generated data figures. ST made illustrations. FM and AB developed and conducted the analysis of the experimental data. All authors contributed to data analysis, discussion, writing, editing, and revisions.

**Competing interests.** At least one of the (co-)authors is a member of the editorial board of *Earth Surface Dynamics*. The peer-review process was guided by an independent editor, and the authors also have no other competing interests to declare.

**Disclaimer.** Publisher's note: Copernicus Publications remains neutral with regard to jurisdictional claims made in the text, published maps, institutional affiliations, or any other geographical representation in this paper. While Copernicus Publications makes every effort to include appropriate place names, the final responsibility lies with the authors.

**Acknowledgements.** Sophie Katzung implemented the first numerical realization of the Poisson model and explored some of the implications of the random walk formulation during an internship with Jens M. Turowski. Fergus McNab was supported by an ERC Consolidator Grant (no. 863490, GyroSCoPe) to Taylor Schildgen. We thank three anonymous reviewers for their constructive comments on previous versions of the manuscript.

**Financial support.** This research has been supported by the European Research Council, H2020 European Research Council (grant no. 863490).

The article processing charges for this open-access publication were covered by the Helmholtz Centre Potsdam – GFZ German Research Centre for Geosciences.

**Review statement.** This paper was edited by Anastasia Piliouras and reviewed by three anonymous referees.

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
