# Peer review of "Width evolution of channel belts as a random walk"

_EGUsphere, 2024_

## Author Comment (AC1)

Rebuttal for "Width evolution of channel belts as a random walk" Jens M. Turowski, Fergus McNab, Aaron Bufe, Stefanie Tofelde

We thank the reviewers for their supportive comments. We have substantially revised the manuscript, especially from Section 3 onwards. Summarizing the main changes:

- We have added further information to the method section to explain the rationale behind choosing the data for comparison and giving some more background information.
- We expanded the results section.
- We have included a table (Table 1) of R2 values for the comparison between the analytical equations and the benchmark stochastic model, as well as for the comparison to data.
- We have added material to the discussion, in particular a paragraph on single vs multiple-thread channels.
- We have expanded the table (now Table 2) in the discussion that gives an overview of the results into a 'cheat sheet' that can be printed off by including the equations.
- We have updated figures, in particular including legends, and adding information and cross links in the figure captions
- We have gone through the entire text and edited for improved clarity and readability.

Below, we reply to each comment in *italics*.

**Reviewer #1**

I read the manuscript by Turowski et al. with great interest. It presents significant efforts to characterize the width evolution of channel belts using a comprehensive approach that integrates analytical solutions, numerical models, flume experiments, and field data. Although well-crafted overall, the manuscript would benefit from improved readability in later sections.

Thanks for the comments. We have revised the manuscript, particularly the later sections, as outlined below.

**Major comments.**

(1) Clarification of river types/planforms that create the channel belt. The manuscript effectively presents analytical solutions in a step-by-step manner that is easy to follow. However, it remains unclear whether these solutions are intended for meandering rivers or braided rivers. This distinction is crucial as the manuscript appears to alternate between discussing belt width evolution in both types of channels. If the authors believe that the differences between these river planforms are not significant, please articulate this argument early in the manuscript and support it with evidence. The described workflow seems more akin to a single-thread channel that widens the belt through a random walk process, raising questions about its applicability to multiple-thread river channels. Clarity on this point in the introduction, methodology, results, and discussion sections would greatly enhance the manuscript's impact.

This is an excellent point. We cannot fully answer it at the moment. The model is set up assuming a single channel. Yet, the experiments of Bufe et al. frequently featured braided channels, but still compare well to the model. We had explicitly discussed this in our previous manuscript on fluvial valleys (Turowski et

al., 2024), and have now included a paragraph summarizing the arguments in section 5.1. We also added some clarifying statements when introducing the model.

(2) Detail and clarity in later sections.

The manuscript maintains excellent readability up to Chapter 3, after which the presentation becomes more abstract and details become sparse. Specifically, Section 3.2 leaves readers uncertain about the types of rivers that produce the discussed floodplains and whether these are confined or unconfined settings. This section would benefit from additional background information. The Results chapter is notably brief, and the figures often include minimal descriptive text, sometimes limited to a single sentence or solely interpretations. While I understand there is much information in the captions, a more thorough description of each figure within the text would help readers follow the narrative more effectively and discern whether there is consistency across different data sets. *We have revised the latter parts of the manuscript, inserting signposts and pointers in most of the*

sections. See replies to reviewer #2's comments for more detail. Figures and figure captions have also been revised.

Specific comments

Abstract: The abstract contains numerous technical terms that may not be accessible to a broader audience, such as "first passage time" and "sediment residence time." Please consider providing brief definitions of these terms within the text constraints.

The abstract has been revised, and we have tried to amend this issue.

Line 9: It's not necessary for floodplains to form exclusively during channel migration; overbank flooding is a predominant factor.

Agreed. We removed the term 'floodplains' from the list.

Line 27-28: The comparison is made between the model and field and flume data. Could you clarify how analytical solutions compare? *The text has been revised.*

Line 28: temporal evolution "of" channel belts – off is missing. *Added, thanks.*

Line 57: braiding river – braided river *Changed.*

Line 334: "structures a distance b from the river" -- "structures located a distance b from the river" *Changed as suggested.*

Line 374: when mentioning these ratios, suggest clarifying these are for unconfined, moderately confined, or highly confined scenarios to provide context for the chosen values. *Changed as suggested*

Section 3.2: This section is too brief to provide a clear understanding of the comparisons being made.

We have amended and rewritten the paragraph to provide some more information on why we chose the three studies for comparison. It is beyond the scope of the paper to fully review and discuss all available data (Pizzuto et al. 2017 provided a recent review). Our main point is to show that the scaling applies to some rivers in conditions that are close to the assumptions underlying the model. We added to section 3.2:

"The controls on sediment ages in natural rivers can be complicated. Depending on the location, sediments may not be only deposited by laterally migrating channels, but also by overbank deposition, tributaries, or other processes such as soil erosion or debris flows. We thus do not expect the sediment age distribution in every river to follow the prediction of our model (eqs. 34-36). To compare to predictions, we picked three channels with published age distribution that feature conditions close to the assumptions of the model: single threat channels undisturbed by processes other than fluvial deposition and erosion (e.g., debris flows), without major tributaries in the study area."

Section 3.3: The experiments concern braided rivers, but it remains unclear whether the random walk model applies to the same river types.

We have not addressed this in this section, but added a paragraph in the discussion to section 5.1. See also reply above.

Results chapter: Please describe each figure in detail before discussing comparisons or discrepancies. This approach will help readers better understand the visuals and their significance. *The results chapter has been considerably edited and expanded.*

Figure 2a: in the legend, Hw = 1, 10, 100, which should likely be Hw/h, I assume. Also, please mention these ratios in the caption and ideally link them with confinement and unconfinement. *Revised.*

Figure 3a: consider using drift distance, since displacement is not mentioned or defined earlier. We used the term 'drift' to refer to the third development phase (square root scaling), and the plots here show all three phases. As such, we changed to 'distance from origin'.

Line 542: Lancaster et al. work is not presented, although its inconsistency with the results is suggested. Please clarify why the three digitized datasets were selected and their differences from Lancaster et al.'s work.

The criteria for selection have been clarified. The Lancaster cases are discussed and interpreted in the last paragraph of section 5.3.

Line 551: Insert "where" before "both channels in confined valleys" *The sentence has been revised.*

Conclusions: The conclusion seems too general and does not succinctly summarize the findings of this study. It reads more like an overview of the study's significance. Please highlight the key findings here. This is right, we believe that the main point of the conclusion is to highlight a study's significance. The abstract summarizes the study. We have expanded the conclusion and included a brief summary of the main results.

**Reviewer #2**

I thoroughly enjoyed reading this manuscript. Turowski et al. present a thorough and detailed exploration into an analytical framework to model the evolution of river channel belts using the statistics of a random walk process. The authors suggest that channel-belts evolve in three phases, a linear expansion, exponential growth, and drift, each with unique statistical properties. The model itself is intuitive and the authors are quite successful in clearly walking the reader through its formulation. The manuscript provides an extremely strong motivator for future work including opportunities for its validation in natural rivers, and its utility in alluvial river restoration and risk quantification. I believe that the model is an important contribution and a paper in this style will be of broad interest to those working within river landscapes. I have one major comment and a few minor comments with the current version.

Thanks for the positive comments!

**Major comments:**

The loose structure of the latter half of the manuscript (section 3 and onwards) could be improved, specifically focusing on grouping similar points of focus and adding more detail. Some examples of how the current framing needs more detail and structure:

Thanks for the comments. We have revised the later sections of the manuscript considerably.

Section 3 includes the set up to the external procedures and datasets used to validate the model. I agree that validation for the model is integral for the success of the paper, but the authors need to flesh out how these datasets/numerical procedures can validate the analytical model. There is no discussion of how success will be evaluated. This section should tell the readers how they should know the analytical model does a good job at representing reality.

We have added introductory sentences to set up the section for all three parts of section 3. Actually, we do not think we do a good job in demonstrating that the model adequately describes reality. The data we use are quite limited and there is a lot that could (and needs to) be done. Partially our decision to leave the comparison to data on a fairly superficial level is due to the scarcity of suitable data. Partially, it is due to the already long and complicated paper. We have picked the two comparisons because they represent scaling relationships that are typical for random walk processes. A positive comparison is therefore a strong indication that the generic model idea of a random walk is a reasonable approach for modelling the dynamics of channel belts. We have now made this rationale explicit in section 3.

The results section (section 4) is currently formatted as a grouping of figures. I would expand this section to walk through what the figures are showing. If section 3 is set up well, this could include things like descriptions of r-squared values between the numerical and analytical results. *We have included R2 values for the different comparisons. The results chapter has been considerably edited and expanded.*

As a reader, there seems like there could be a distinction between the validation of the model and the application of the model. To explain what I mean by this, the numerical simulations (section 3.1) and the examination of the Bufe 2016, 2019 experiments (section 3.3) are great examples of opportunities to validate the model. On the other hand, the sediment residence times seems like a natural application of the model, using it as a tool for geochemistry, not just geomorphology. In the current version of the manuscript, these components are grouped together in shared sections. I think some clarity could be gained by separating the ideas of validation and application in the text. One could even section out the paper to have a specific validation section, then a specific application section, which could help break up the results into a natural structure and motivate expanding section 3.

We think that we understand the point the reviewer makes here, and partially agree. Yet, given that we do not make a comprehensive comparison to data in the study, and all comparisons are imperfect, we think the distinction of validation and application would overemphasize this explorative part of the manuscript. We have clarified the reasons for picking the data sets we use in section 3 (see above).

**Minor comments:**

The introduction could use a stronger statement as to why predictions and theory for the evolution channel-belts are important. The authors could lean on the value of the predictive model specifically. A model connecting channel-belt width through time to hydraulic variables has direct applications in quantifying floodplain risk and management (forward model), as well as to back out hydraulic properties (or even relative migration timescales) from direct measurements of channel-belt widths in fluvial strata (inverse model). There are a couple places in the text (outside of the introduction) where this type of implication/context is given in more detail. I would suggest moving those to the set-up of the paper. *We have added a few sentence at the end of the second-to-last paragraph:*

"Equations relating the growth evolution of channel belts and valleys to the hydraulic conditions in the channel are currently not available. Yet, they could be useful in diverse topics. For example, they could be used as forward models for making predictions relating to flood hazard assessment and stream restoration, or as inverse models to obtain paleo-hydraulic conditions from fluvial stratigraphy and depositional sequences. Further, they could provide a framework to interpret data from natural rivers with regard to nutrient cycling, channel-floodplain interactions, or ecology."

Be explicit about planform applicability. The fact that it works for the Bufe experimental data suggests that the model is widely applicable, but it would be nice throughout the model set up do discuss explicitly that planform differences are an important consideration. Intuition suggests to me at least, that delta\_T (delta\_t) should be different depending on if a river has a single, or more than one active thread. We have added a paragraph in section 5.1 on this topic, summarizing a discussion from our previous paper on valley width (Turowski et al., 2024). See replies to reviewer #1 for more details.

**Line-by line comments:**

Eqs (7) and (8) – I may be unfamiliar with standard notation, but the left side of equations could be more descriptive. Instead of PDF\_exponential, they could be PDF\_TW and PDF\_dx respectively. The functional form of the PDF is anyhow seen on the right side. Would help connect the probability functions to what they represent in the equation (not just vis-à-vis the text). *Changed as suggested.*

L227: Keep language consistent around unconfined and confined channel belt. Should be formalized in the set up. "Unconfined plane" is a new way of referring to this. *Changed to* "...the evolution of an unconfined channel belt."

Section 2.3.3: The section is clear. The derivation of c, while cool to see how it falls out of the model, could be moved to a supplement to simplify text.

We think that the value of c is a key result that should be part of the main manuscript. First, it allows for a direct comparison with the stochastic numerical model, without the need of any fits. Second, it relates the effective switching time scale to the rate constant. Third, it highlights that the steady state width corresponds to the standard deviation of the width distribution. Fourth, it fixes one of the constants in the model, leaving a single calibration parameter. We considered to move the derivation into its own subsection, but it seems too short to warrant this. As such, we have not changed anything.

L311: "Knowledge of this distance..." exactly the type of language that could be rolled into the motivation to the problem (section 1). Having this already known to the reader would mean you could jump right into the derivation after the topic sentence.

The derivation is a consequence of the random walk set up of the model, and thereby contingent on the assumption used to build it. The statement referred to here are specific to the first passage distributions. We think it would be difficult to include such a statement in the introduction in its current form, as it would need an introduction to stochastic models and random walks to make sense. We have therefore decided to keep the explanation where it was. We have revised the final sentences in the second-to-last paragraph of the introduction to connect the quantitative description of channel belt evolution to its potential uses: "Equations relating the growth evolution of channel belts and valleys to the hydraulic conditions in the channel are currently not available. Yet, they could be useful in diverse topics. For example, they could be used as forward models for making predictions relating to flood hazard assessment and stream restoration, or as inverse models to obtain paleo-hydraulic conditions from fluvial stratigraphy and depositional sequences. Further, they could provide a framework to interpret data from natural rivers with regard to nutrient cycling, channel-floodplain interactions, or ecology."

L:321 should the "(eq. 15)" be "(eq. 20)?" *Corrected.*

L376: "limiting the channel-belt width to the steady-state width by adjusting the other side of the valley." This is an interesting assumption to keep examining because there are so many unknowns at play. For a numerical (or analytical) model, regardless if vegetation recolonizes the other side of the channel-belt, in terms of its evolution, it's still part of the same channel-belt even though it is statistically unlikely that it would be visited by the channel. This remains true for a natural channel-belt between avulsion events. I don't have much intuition outside of unconfined channel belts, but I would not take the channel belt width preservation to be a widely valid assumption.

Fair enough. The mathematical treatment is interesting for several reasons. First, it yields the value of c. Second, it may be much more relevant for incising valleys than for unconfined channel belts. We have added an appendix with some equations that could be used for landscape evolution models. Otherwise, we have not changed anything.

L395: No topic sentence as to why the authors are leveraging this experimental dataset. The opening paragraph to the section simply states that the "results are compared to two separate types of data... (ii) the temporal evolution...in the experiments of Bufe et al..." I know implicitly that it's to validate the model against scarce empirical data, but the authors never explicitly state as much. This relates to my major comment.

We added a few sentences at the start of the section, explaining our aims.

"We validate the model using experimental data of Bufe et al. (2016a) and Bufe et al. (2019). Primarily, we seek evidence for the drift phase, i.e., a square root scaling of the average channel belt width with time in the later parts of the experiments. This would be a strong confirmation that channel belt development can be described as a random walk."

Results section: Check figure numbers throughout this section. *Checked, thank you.*

L416: This should be Fig. 7 not Fig. 6. *Corrected.*

Fig. 7: It's unclear to me why you are only fitting the drift phase to the data? Given the long experimental run times relative to the mobility, shouldn't you be able to recognize all three phases of growth and evolution?

This is an interesting question and we have spent quite some time exploring it. We have opted to just show the drift phase because the model is not fully constrained from the experimental data. That is, there is one free parameter, the unconfined floodbelt width, which can be chosen arbitrarily. The data does not currently give any constraints on this. As a result, plotting the exponential phase does not provide any further information. This had been stated in the last paragraph of section 5.1. We moved this statement to the results section now.

L554-560: If you aren't going to discuss q\_L, and I agree that it's not necessary, you can remove this text and simply set up a topic sentence (and section heading) about k. *Agreed, we removed the first couple sentences.*

L571: The switch between the statement touching on directly estimating k, and an empirical model for  $\lambda$  is sudden and not set up well. This paragraph could use a topic sentence for what this derivation is arriving at.

Agreed, we added a connecting statement: "Our model is constructed at the reach scale of the channel and does not include detailed descriptions of fluvial processes. Yet, it should be possible to relate it to process-based models. Here, we make a tentative relation to models of meandering channels, which are available at different degrees of complexity (e.g., Edwards & Smith, 2002; Ikeda et al., 1981)."

Section 5.5: Would the authors make the suggestion that their simplifying assumptions of independent cross-sections wouldn't lead to fruitful results when examining 2d planform changes? The reviewer's statement is not totally clear to us. Our expectation is formulated in the last paragraph of the section: we expect that the functional relationships are the same, but dimensionless constants differ (slightly?) when channel-longitudinal effects are included. Whether this expectation is correct and

whether the model as formulated in the present paper can describe real river systems needs to be tested with dedicated data that is not available at the moment. We have slightly changed the introductory sentence to avoid suggesting that we do not believe our own model.

---

## Referee Report (RR1)

Thanks again for the opportunity to review this manuscript once more. In short, I think the authors did a great job at addressing my major concerns, and I believe that the paper is suitable for publication. Specifically:

The introductory paragraph for Section 3 (L403) is successful in setting expectations for the results sections and helps orient the reader. The new structure of the "Testing the Stochastic Poisson Model" section makes the following the results section easier. I appreciate the inclusion of $R^2$ values throughout, and it's easy to trace the story from figure to figure. Finally, I appreciate the new paragraph in the discussion (L646) about the applicability across planforms.

My only comment, and I leave this to the discretion of the authors, is that table 1 may not be needed; especially given that this information is repeated in line with the text. It's a very minor comment, and the table may actually be nice to have within the final print/web formatted version.

---

## Author Response (AR2)

**Rebuttal**

We thank the reviewers and editors for their work. Reviewer #2 had some minor comments on the last version of the manuscript, which we have addressed in the revised version. We reply to all of the comments below in *italics*.

Reviewer #2
This manuscript presented a theoretical work on modeling the width evolution of channel belts as a Poisson process. Specifically, the channel path is modeled as a 1D random walk with a constant rate related to channel hydraulic parameters. Three growth phases are identified via linear, exponential, and drift phases. Bounds of the channel belt are also modeled via the law of the iterated logarithm, which has implications for flood hazard monitoring. Another novel finding is that a floodplain sediment age distribution proxy was also derived, including two practical approximations on top of the full numerical solution. The age distribution model was tested well against measurements of natural and experimental rivers. Overall, I think this paper is well-written and organized. I have read an earlier version of the manuscript and was not able to provide reviews on time (I apologize), but most of my comments (and the other reviewers') are already addressed. I appreciate the nicely made figures, referencing table for the equations, and list of variable names. I think the novelty of this manuscript is about constraining the rate parameters in the various distributions using channel width, valley height, and lateral transport capacity. All three parameters have physical meaning and can be measured in the field. At the same time, each of the three parameters has a range of complexity. For example, a range of hillslope processes are embedded in lateral transport capacity and valley height, and the channel width is the timeless hydraulic geometry problem in fluvial geomorphology. I think the thought process and the models provided here set a nice foundation that hopefully (will) inspire a series of future works on the more holistic understanding of how sediment transport processes in rivers shape its channel and channel corridor.
*Thanks for the supportive comments!*

I only have a few minor comments, hoping to improve the manuscript for Esurf's readers.
Comments:
I'd suggest a quick clarification about the steady state (in terms of mass balance): Is the width reaching a dynamic constant value, the bed slope, or both, and over what timescale?
*Here, we are only concerned about the steady width of the channel belt. We treat both of the cases where belt width is steady and the entire belt drifts laterally (section 2.3.3) and a stochastically increasing belt width over time (section 2.3.2).*
*In all the derivations, we assume that the lateral transport capacity can be treated as a constant that depends on boundary conditions including water discharge, upstream sediment supply and granulometry. The experiments of Bufe et al. (2019), which were used to develop the concept of the lateral transport capacity, indicate that the concept encompasses autogenic variability within the channel geometry.*
*We added a sentence in section 2.1:*
*"The lateral transport capacity can be treated as a constant for a given set of boundary conditions including water discharge, upstream sediment supply, and granulometry (Bufe et al., 2019)."*

Meandering rivers can develop confined valleys through autogenic processes (e.g., Limaye et al., 2013, often in lowlands). Does the model distinguish the cause of confinement?
*The cause of confinement is not relevant for the evolution equations. In autogenically confined channels, the lateral migration speed v may depend on time. We do not take this into account.*

I'd suggest labeling W0 and Wv in Fig.1.
*We have updated the figure.*
I would also add a half-sentence around L156 stating that Wv is also known as confined channel belt width OR any means to more explicitly explain the difference between confined channel belt width and steady-state valley floor width beyond the context of Turowski et al., 2024.
*In the present paper, we do not explicitly account for uplift and the different notation is meant to reflect this. That is, Wv refers to the steady state width including the effect of uplift, while we use W0 to denote a steady state width unaffected by uplift.*
*We added:*
*"The valley-floor width WV is distinguished from the confined channel belt width by explicitly accounting for the effects of uplift and lateral sediment supply."*

Also, is it fair to say that properties that impact the erodibility of the valley wall, such as lithology, are embedded in qH?
*Valley wall erodibility affects the lateral migration speed and can be captured by the difference between speed V in the floodplain and v when migrating beyond it (see eq. 15 and following). The erodibility affects the transient approach to steady state and the drift (as quantified in later parts of the manuscript), but not the steady state width. There is a more elaborate discussion on this issue in our previous paper (Turowski et al. 2024).*
*It can be expected that qH is affected by bedrock lithology, but the relationships are not clear at the moment.*

Fig 1b-d is missing the subscript 0 on the width.
*We have updated the figure.*

L 598-600, the model presented here, can possibly be applied to a broader range of rivers. See Dong and Goudge, 2022, which provided a relationship between river planform pattern and channel belt width.
*We had not been aware of the Dong and Goudge paper, thanks for pointing it out! We now cite it in the introduction and discussion.*